# Experimental evidence that uniformly white sclera enhances the visibility of eye-gaze direction in humans and chimpanzees

**Fumihiro Kano**[1,2,3]*, **Yuri Kawaguchi**[4,5,6], **Yeow Hanling**[2]

[1]Centre for the Advanced Study of Collective Behaviour, University of Konstanz, Konstanz, Germany; [2]Kumamoto Sanctuary, Kyoto University, Kumamoto, Japan; [3]Max Planck Institute of Animal Behavior, Radolfzell, Germany; [4]Messerli Research Institute, University of Veterinary Medicine Vienna, Vienna, Austria; [5]Japan Society for the Promotion of Science (JSPS), Tokyo, Japan; [6]Primate Research Institute, Kyoto University, Inuyama, Japan

**Abstract** Hallmark social activities of humans, such as cooperation and cultural learning, involve eye-gaze signaling through joint attentional interaction and ostensive communication. The gaze-signaling and related cooperative-eye hypotheses posit that humans evolved unique external eye morphologies, including uniformly white sclera (the whites of the eye), to enhance the visibility of eye-gaze for conspecifics. However, experimental evidence is still lacking. This study tested the ability of human and chimpanzee participants to discriminate the eye-gaze directions of human and chimpanzee images in computerized tasks. We varied the level of brightness and size in the stimulus images to examine the robustness of the eye-gaze directional signal against simulated shading and distancing. We found that both humans and chimpanzees discriminated eye-gaze directions of humans better than those of chimpanzees, particularly in visually challenging conditions. Also, participants of both species discriminated the eye-gaze directions of chimpanzees better when the contrast polarity of the chimpanzee eye was reversed compared to when it was normal; namely, when the chimpanzee eye has human-like white sclera and a darker iris. Uniform whiteness in the sclera thus facilitates the visibility of eye-gaze direction even across species. Our findings thus support but also critically update the central premises of the gaze-signaling hypothesis.

**\*For correspondence:**
fkano@ab.mpg.de

**Competing interest:** The authors declare that no competing interests exist.

## Editor's evaluation

The study by Kano et al. is notable for being the first to adopt a comparative experimental approach, testing both humans and chimpanzees in a cross- and within-species design, demonstrating that uniformly white sclera enhances eye-gaze discrimination in both species. Crucially, their results support the gaze-signaling hypotheses for the evolution of particular features of the human eye but further suggest that uniformly white sclera are critical for eye-gaze discrimination when visibility conditions are poor.

## Introduction

In humans, eye-gaze is employed in critical ways for conspecific communication during social activities such as cooperation, teaching, and language learning (*Csibra and Gergely, 2009*; *Tomasello et al., 2005*). Humans excel at detecting another's direct gaze and following another's gaze directions from

**eLife digest** From an early age, we are able to detect the direction others are looking in (known as eye-gaze) and make eye contact with each other to communicate. The front of the human eye has a large white area known as the sclera that surrounds the darker colored iris and pupil in the center.

Compared to us, chimpanzees and other nonhuman great apes have sclerae that are much darker in color or at least not as uniformly white as human eyes. Some researchers believe that the white sclera of the human eye may have evolved to make it easier for other individuals to detect the direction of our gaze. However, there is a lack of experimental evidence as to whether white sclerae actually helps humans to distinguish the direction of eye-gaze.

Here, Kano, Kawaguchi and Yeow presented human and chimpanzee participants with images of other humans and chimpanzees on a computer screen and asked them to indicate the direction of eye-gaze in each image. The experiments found that both humans and chimpanzees were better able to discriminate the directions of eye-gaze from the images of humans than those of chimpanzees, particularly when the images were smaller or more shaded. Moreover, artificially altering the eyes in the chimpanzee images so that they were more human-like – that is, had a light-colored sclera and a darker iris – enabled both humans and chimpanzees to better discriminate the eye-gaze directions of the chimpanzees.

Kano, Kawaguchi and Yeow's findings indicate that white sclerae do indeed help both humans and chimpanzees to discriminate the direction of eye-gaze, even though only humans have white sclerae. This is likely because humans use eye-gaze in key social activities (including learning languages, coordinating to complete complex tasks and transmitting cultural information), indicating that white sclerae may have evolved to enhance human-specific communication. To learn more about this type of communication, future research could focus on finding out when white sclerae first evolved.

early infancy (*Farroni et al., 2002*; *Farroni et al., 2004*). Moreover, during social interactions, humans exchange communicative intentions with one another via eye contact (*Senju and Johnson, 2009*; *Kleinke, 1986*), and an experience of being watched by another affects one's reputational concerns (*Bateson et al., 2006*; *Engelmann et al., 2012*). Comparative studies have found that closely related species such as nonhuman great apes also excel at detecting another's direct gaze (*Myowa-Yamakoshi et al., 2003*) and following another's gaze directions (*Bräuer et al., 2005*; *Okamoto et al., 2002*), and adjust their gaze-following behaviors flexibly and cognitively (*Kano and Call, 2014*; *Okamoto-Barth et al., 2007*; *Povinelli and Eddy, 1996*; *Tomasello et al., 1999*). Moreover, they make eye contact with other individuals at critical moments of social interactions, such as when requesting food from an experimenter (*Gomez, 1996*) and initiating affiliative social interactions with conspecifics (*Heesen et al., 2021*; *Yamagiwa, 1992*). However, several potential differences might exist between humans and nonhuman apes in their use and interpretation of gaze behaviors. For example, in the gaze-following/cueing experiments, while humans primarily relied on another's eye-directional cues, nonhuman apes relied on head-directional rather than eye-directional cues (*Tomasello et al., 2007*; *Tomonaga, 2007*, but see *Povinelli and Eddy, 1996*; *Deaner and Platt, 2003*). In general, reliance on eye-directional rather than head-directional cues indicates one's inclination/ability to identify a specific spot that another's foveae are directing at, rather than a large area that another's field of view covers. Thus, the former strategy leads to efficient identification of the interaction partners' focused objects during joint attentional interaction and ostensive communication. Relatedly, a previous study observed that, while both human infants and nonhuman apes alternately looked at both the interaction partner's face and the focused object during joint-attentional interactions, nonhuman apes made only brief looks to the interaction partner's face while human infants made more long-lasting looks (*Carpenter and Tomasello, 1995*). Furthermore, in a communicative gaze-following task, while human infants responded to an agent's eye contact as if they interpreted it as an ostensive signal preceding referential information (*Senju and Csibra, 2008*; *Okumura et al., 2020*; but see *Gredebäck et al., 2018*), great apes did not show any evidence supporting this understanding (*Kano et al., 2018*). These potential differences between human and nonhuman apes might be partly related to differences in their sociocognitive skills but also humans' specialization to eye-gaze signals in conspecific communication.

Relatedly, one influential hypothesis, the gaze-signaling hypothesis (*Kobayashi and Kohshima, 2001*; *Kobayashi and Kohshima, 1997*), proposes that humans have evolved special morphological features in the eye, including uniform whiteness in the exposed sclera, to enhance the visibility of eye-gaze directions and thereby help conspecifics communicate via eye-gaze without much attentional effort. This hypothesis is based on comparative analyses showing that (1) uniformly white sclera is a unique feature of the human eye among primates and that (2) the human eye is horizontally more elongated, and the human sclera is horizontally more exposed compared to other primates' eyes (*Kobayashi and Kohshima, 2001*; *Kobayashi and Kohshima, 1997*). The cooperative-eye hypothesis (*Tomasello et al., 2007*) extended this gaze-signaling hypothesis based on the results from their gaze-following study, proposing that humans evolved these morphological features as well as a special sensitivity to these features to facilitate joint-attentional and communicative interactions in a cooperative context. Others discussed that humans' white sclera not only signals gaze direction but also emotional cues in combination with fine musculatures around the eyes (*Whalen et al., 2004*; *Baron-Cohen et al., 2001*; *Jessen and Grossmann, 2014*), and also emotional and health-status cues by its color variations (*Provine et al., 2013a*; *Provine et al., 2013b*). Moreover, typically developing human adults and children show basic preferences for an animal agent having white sclera (*Segal et al., 2016*).

Despite the widespread popularity of the gaze-signaling and related cooperative-eye hypotheses in the literature, these hypotheses have been severely challenged by recent quantitative morphological studies showing that certain eye features of humans are not necessarily unique among great ape species (*Mayhew and Gómez, 2015*; *Perea-García et al., 2019*; *Caspar et al., 2021*; *Mearing and Koops, 2021*; *Kano et al., 2021*). Accordingly, there is currently a heated debate over whether the external eye morphology of humans has any communicative function. Specifically, those studies have shown that (1) the sclera of humans is not necessarily more exposed than that of other great ape species (*Mayhew and Gómez, 2015*; *Caspar et al., 2021*; *Kano et al., 2021*), that (2) the color contrast/difference between the iris and the sclera is similar between humans and other great apes (*Perea-García et al., 2019*; *Caspar et al., 2021*; *Mearing and Koops, 2021*; *Kano et al., 2021*), and that (3) there is substantial individual variation in the extent to which sclera is unpigmented among some nonhuman ape species (*Mayhew and Gómez, 2015*; *Caspar et al., 2021*; *Mearing and Koops, 2021*; *Kano et al., 2021*; *Perea García, 2016*). Given these new findings, one recent study suggested that one (and possibly only) eye feature that both distinguishes humans from other great apes and contributes to the visibility of eye-gaze direction is *uniform* whiteness in humans' exposed sclera; i.e., depigmentation all the way from the iris edge to the eye corners (*Kano et al., 2021*). This previous study used image analysis and computer vision techniques and identified that uniformly white sclera characterizes clear visibilities of both iris and eye-outline edges – the two essential features that contribute to the visibility of eye-gaze directions (see *Appendix 3—figure 1* for the illustration of this aspect). More specifically, while the iris is highly visible in all great ape species, the visibility of eye-outline edges is limited in nonhuman apes because most nonhuman ape individuals have darker or more graded/patchy sclera colors compared to humans' sclera colors, which more easily blend into the adjacent skin/hair colors. On the other hand, nearly all human individuals have uniformly white sclera, which is clearly distinguished from the adjacent skin/hair colors around the eyes (irrespective of the skin color variations) even when the eyes are viewed in visually challenging conditions (e.g., shading, distancing). However, experimental evidence is lacking to support these findings.

Previously, several experimental studies demonstrated that humans' white sclera facilitates gaze perception at least in human participants. *Ricciardelli et al., 2000* tested human participants in a gaze-discrimination task and found that reversing the contrast polarity of human eye images makes the judgment of gaze direction less accurate. Specifically, the gaze directions were more accurately judged in eyes having a positive (normal) contrast polarity (white sclera with a darker iris) than those having a negative (reversed) contrast polarity (black sclera with a bright iris), even though these two eye images were composed of the same colors. Ricciardelli et al. thus suggested that humans have perceptual expertise in positive contrast polarity of eyes (see *Itier et al., 2006*; *Yoshizaki and Kato, 2011* for related results). Also, *Yorzinski and Miller, 2020* tested human participants in a gaze-discrimination task in which the sclera colors of human faces were manipulated to be either conspicuous (white or lighter than the iris color) or inconspicuous (similar or darker than the iris color); human participants detected the faces with conspicuous sclera colors faster and more accurately. Yorzinski

and Miller thus suggested that humans' white sclera facilitates their gaze perception. Importantly, although these two studies found similar results, it remains unclear whether humans' white sclera facilitates their gaze perception by its perceptual properties per se or by human participants' perceptual expertise in the positive contrast polarity of human eyes. To answer this question, a comparative approach is helpful.

*Tomonaga and Imura, 2010* tested whether a chimpanzee could be trained to discriminate eye-gaze directions of human facial images, and in one of their experimental conditions, they replicated the results from *Ricciardelli et al., 2000*. Specifically, they showed that the chimpanzee's task performance dropped when the contrast polarity of the eyes was reversed in the stimulus human faces. Unfortunately, however, the conclusion from this result is severely limited because the chimpanzee was trained to discriminate the gaze directions of the human positive eyes (positive contrast polarity) before being tested in the trials presenting the human negative eyes (reversed contrast polarity). Therefore, it remains unclear whether the chimpanzee's performance was affected by the change in the eye contrast polarity or merely the change in the trained color pattern. Critically, no previous study adopted a fully crossed design for species comparison to examine the effect of sclera colors on primate gaze perception; namely, a study design that presents the stimuli of both species to the participants of both species. Chimpanzees are a suitable species for this design because their eyes have relatively uniformly dark sclera and a bright iris, namely, having a negative contrast polarity opposite to the positive contrast polarity that human eyes have. Therefore, in a fully crossed design with chimpanzees and humans, one can test whether the uniformly white sclera of human positive (normal) eyes and chimpanzee positive (reversed) eyes facilitates the gaze-discrimination performances of both chimpanzee and human participants. Such a question would clarify whether the uniformly white sclera has a perceptual advantage independently from perceptual advantages of other eye features or the participants' perceptual expertise in a certain eye contrast polarity (as we will detail below).

Consequently, this study tested both humans and chimpanzees on their ability to discriminate the gaze direction of both chimpanzee and human faces with the eyes manipulated to have both normal and reversed contrast polarities. The contrast polarity of both species' eyes (only the eyeball regions) was reversed by inverting the lightness (or grayscale) values of the eyeball images in the faces (*Figure 1A* and *Appendix 3—figure 3*). This manipulation created artificial white sclera in chimpanzee eyes and artificial dark sclera in human eyes, while not affecting the iris-sclera (or pupil-iris) color differences in each species' eyes. The task for human and chimpanzee participants was to discriminate the gaze direction of human and chimpanzee eye images on a computer monitor (either in a keypress or visual search task; *Figure 1B*). Our experimental procedures followed a previous study adopting training procedures (*Tomonaga and Imura, 2010*) instead of spontaneous gaze-following/cueing procedures for chimpanzees (*Tomasello et al., 2007*; *Tomonaga, 2007*) because, while the former study successfully trained a chimpanzee to discriminate eye-gaze directions of a stimulus (human) face, the latter studies reported that chimpanzees predominantly rely on head-directional cues but not eye-gaze directional cues in the spontaneous tasks. Also, as the strength of the visual signal generally depends on its degradation caused by natural noises (*Endler, 1990*), we tested the robustness of eye-gaze signals against shading and distancing (*Figure 1A*), namely the simulated visual conditions where stimulus eyes were presented darker (in shadows) and smaller (in the distance), following a previous study on great ape eye color (*Kano et al., 2021*).

We developed five sets of hypotheses and predictions (*Table 1*). H1 is our key hypothesis, which posits the perceptual advantage of uniformly white sclera (*Kobayashi and Kohshima, 2001*; *Kobayashi and Kohshima, 1997*; *Kano et al., 2021*) and thus predicted increased performance for both human and chimpanzee positive eyes in participants of both species. H2 posits the perceptual advantage of the iris-sclera color difference. As the iris-sclera color difference (note the difference between the contrast and difference measures employed by previous studies; *Perea-García et al., 2019*; *Caspar et al., 2021*; *Mearing and Koops, 2021*; *Kano et al., 2021*) was similar between chimpanzees and humans in general (*Kano et al., 2021*) and did not differ between our chimpanzee and human stimuli with both positive and negative eyes (see *Appendix 3—figure 2* for the quantitative evaluation of our stimuli), we predicted no performance difference between the stimulus types in participants of both species. H3 posits the perceptual advantage of the horizontally elongated shape. As human eyes were horizontally longer than chimpanzee eyes in general (*Mayhew and Gómez, 2015*; *Caspar et al., 2021*; *Kano et al., 2021*) and also in our stimulus set (*Appendix 3—figure 2*), we predicted

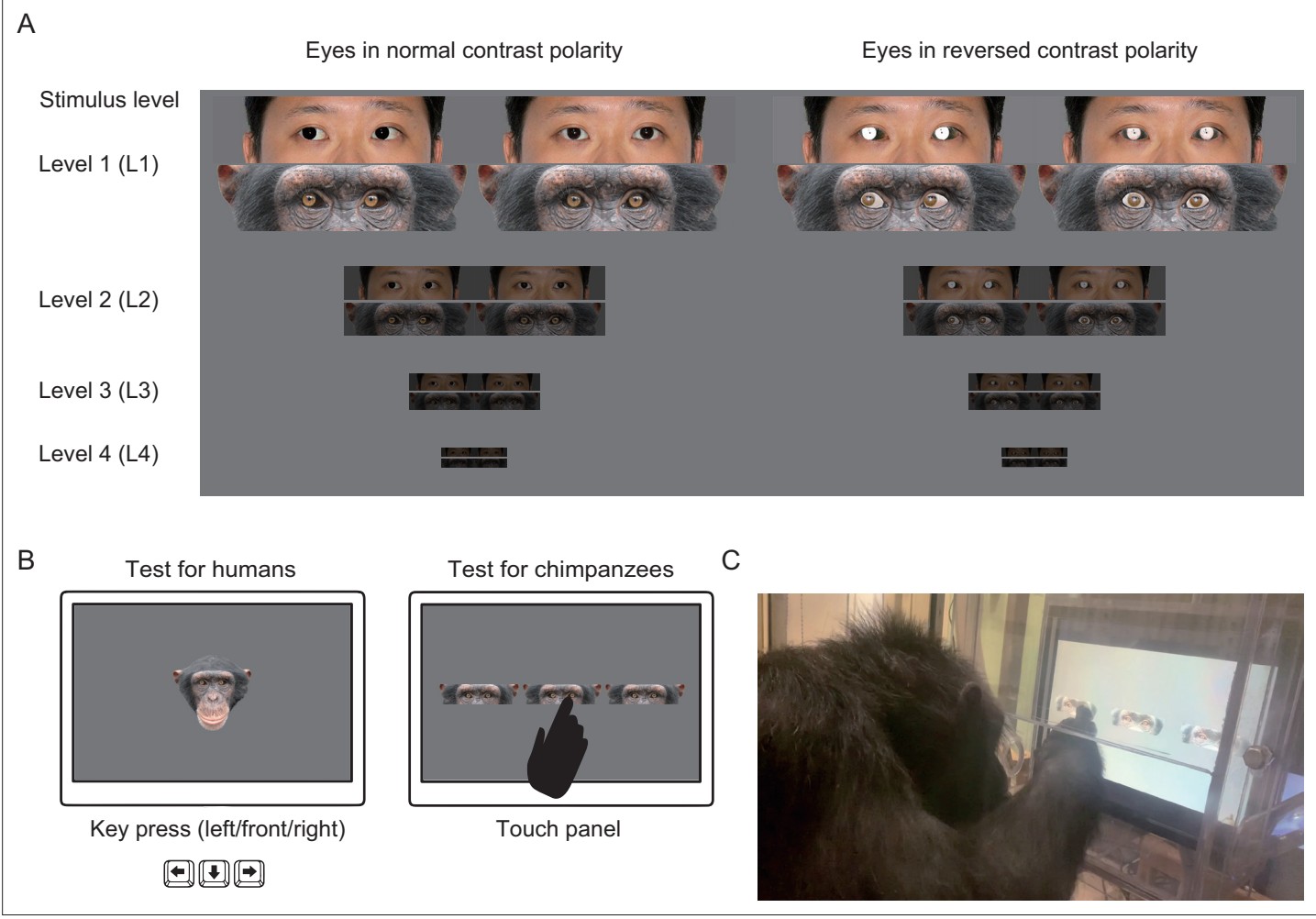

**Figure 1.** Experimental stimuli and procedures. (**A**) Experimental stimuli used in this study. Stimuli consisted of chimpanzee and human eye images with direct and averted (20°) gaze in normal and reversed contrast polarities. The stimulus levels varied according to the brightness and size of stimuli (L1–4), with L1 being the brightest and largest and L4 being the most shaded and smallest. Permission was obtained to publish the human image (this image was only for presentation purposes; not used in this study but edited following the methods used in this study; see *Egger et al., 2011* for the stimuli used in the study). (**B**) Schematics for the tests with human and chimpanzee participants. Participants of both species were presented with the stimuli of both species. In each trial, human participants indicated the gaze direction of each stimulus face by a keypress (left/front/right), and chimpanzee participants indicated the averted gaze face among the two direct gaze faces by a touch response. (**C**) Experimental setup for chimpanzees.

increased performance for the human stimuli (with both positive and negative eyes) in participants of both species. H4 posits the participants' perceptual expertise in the contrast polarity of own-species eyes (*Ricciardelli et al., 2000*) and thus predicted increased performance for the positive eyes of both species in human participants and the negative eyes of both species in chimpanzee participants. H5 addressed a general possibility arising from our manipulation of both species' eye colors, namely, that participants of both species perform more poorly in such artificial conditions, and thus predicted increased performance for the human positive eyes and the chimpanzee negative eyes in participants of both species. Related to this last hypothesis, our chimpanzee participants had extensive experiences in interacting with both conspecifics and humans from youth and thus were familiar with both species' eyes. We also ensured that our human participants had a minimum of a few months (to decades) of experiences in interacting with chimpanzees and thus were familiar with both species' eyes.

## Results

In Study 1, we tested 25 adult human participants (14 females, 11 males) in two experiments. Experiment 1 presented participants with the stimuli of both humans and chimpanzees with normal contrast

**Table 1.** Hypotheses predicting the effect of eye color and shape on the visibility of eye-gaze in chimpanzee and human participants.

| Hypothesis | Assumption | Participant | Eyes with normal contrast polarity | | Eyes with reversed contrast polarity | |
|---|---|---|---|---|---|---|
| | | | Human eye | Chimpanzee eye | Human eye | Chimpanzee eye |
| H1 | Perceptual advantage of uniformly white sclera | Humans | + | − | − | + |
| | | Chimpanzees | + | − | − | + |
| H2 | Perceptual advantage of the iris-sclera color difference | Humans | ~ | ~ | ~ | ~ |
| | | Chimpanzees | ~ | ~ | ~ | ~ |
| H3 | Perceptual advantage of the horizontally elongated eye shape | Humans | + | − | + | − |
| | | Chimpanzees | + | − | + | − |
| H4 | Perceptual expertise in the contrast polarity of own-species eyes | Humans | + | − | − | + |
| | | Chimpanzees | − | +* | + | − |
| H5 | Perceptual expertise in the normal contrast polarity of eyes | Humans | + | + | − | − |
| | | Chimpanzees | + | + | − | − |

*+ denotes a higher performance than −, and ~ denotes a similar performance in within-species (row-wise) comparisons.

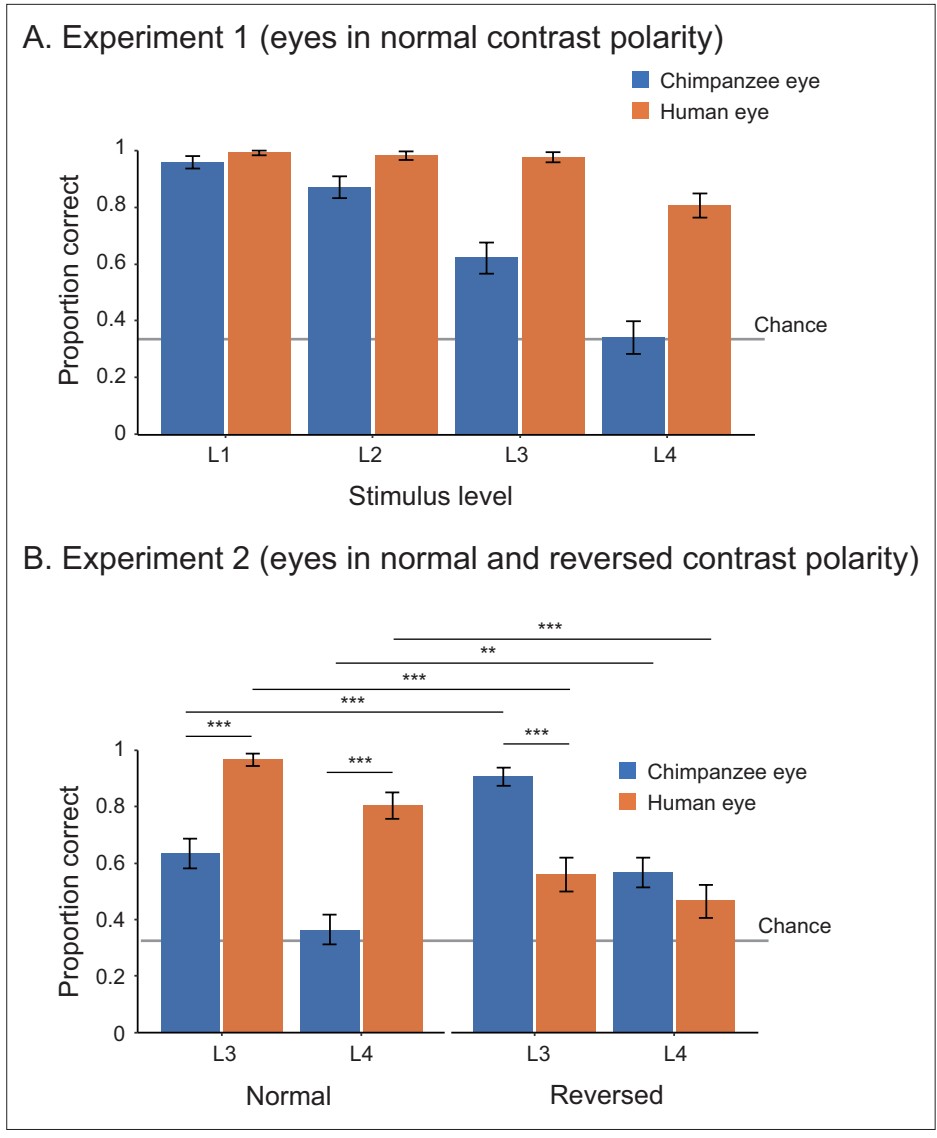

**Figure 2.** Performance of human participants in Study 1, represented as mean proportion correct in experiments 1 (**A**) and 2 (**B**). L1–4 indicate the stimulus level, with L1 being the brightest and largest and L4 being the most shaded and smallest. Error bars are 95% confidence intervals based on a nonparametric bootstrap. Asterisks indicate significance in the post-hoc models ran for Experiment 2 (***p<0.001, **p<0.01).

polarity at four stimulus levels (L1–4) varying in size and brightness. We tested the effect of stimulus level and species on participants' correct responses (correct, incorrect) in a binomial generalized linear mixed model (GLMM; see *Appendix 1—table 1* for the formulas) and found that human participants performed better in trials presenting the human stimuli than those presenting the chimpanzee stimuli ($\chi^2$ = 37.08, df = 1, p<10$^{-8}$). We also found that their performance was worse in trials presenting smaller and more shaded stimuli ($\chi^2$ = 45.85, df = 3, p<10$^{-9}$; see *Appendix 1—table 2* for the full GLMM results; *Figure 2A*). Experiment 2 presented the same participants with the stimuli of both species with both normal and reversed contrast polarities at stimulus levels L3 and L4. We tested the effect of stimulus level, species, and contrast polarity on participants' correct responses in GLMM and found a significant three-way interaction effect between these factors (*Figure 2B*; $\chi^2$ = 17.57, df = 1, p<10$^{-4}$; also see *Appendix 1—table 2*). We then performed simple effects tests to examine the observed interaction effect further (*Figure 2B* and *Appendix 1—table 2*). Critically, we found that human participants performed better in trials presenting the positive (reversed) eyes of chimpanzees (i.e., the white sclera and a darker iris) than those presenting the negative (normal) eyes of

chimpanzees (i.e., the dark sclera and a bright iris), while they performed worse in trials presenting the negative (reversed) eyes of humans than those presenting the positive (normal) eyes of humans.

In Study 2, we began with training 10 chimpanzees. Only three (Natsuki, Hatsuka, Pendesa) passed all the required training and subsequently participated in test phases (see *Appendix 2—table 1* for details about participants; also see *Appendix 4—table 1* and *Figure 1* for the number and performance of training sessions for each chimpanzee). Study 2 involved two experiments. Experiment 1 tested those three chimpanzees and presented them with the human and chimpanzee eye images with normal contrast polarity. To test chimpanzees at stimulus levels higher than L1 (i.e., smaller and more shaded), we gradually incremented the stimulus level (by 0.5) across sessions when individuals showed high performance in target trials presenting a given stimulus level (above 85% in two successive sessions). The test phase was defined as the sessions presenting stimulus level higher than (or equal to) L2.5, given that we observed clear performance differences between stimulus species at stimulus levels higher than L2 in Study 1 (see *Appendix 4—table 3* for the number of sessions in the pre-test and test phases). We tested chimpanzees' correct responses during the test phase across repeated sessions at the individual level (with the $\alpha$ level corrected for the number of individuals in the Bonferroni correction, $\alpha = 0.05/3$). Each chimpanzee completed a minimum of 20 test sessions. To avoid the ceiling effect, we incremented the stimulus level also during the test phase when chimpanzees showed high performance in target trials based on the same criteria. We tested the effect of stimulus species in binomial GLMM on each chimpanzee's correct responses (correct, incorrect) during the test phase and found that all three chimpanzees performed significantly better for the human stimuli than the chimpanzee stimuli (Natsuki: $\chi^2 = 8.28$, df = 1, p=0.004; Hatsuka: $\chi^2 = 9.50$, df = 1, p=0.002; Pendesa: $\chi^2 = 21.94$, df = 1, p<10$^{-5}$; also see *Appendix 1—table 2*).

Experiment 2 tested two (Natsuki and Hatsuka) out of the three chimpanzees. Pendesa was dropped from this experiment because she took about twice as many training sessions as the other two chimpanzees (*Appendix 4—table 2* and *Figure 1*). Experiment 2 presented them with eye stimuli having reversed contrast polarity in the first test phase (Test B) and then eye stimuli having normal contrast polarity in the next test phase (Test A2); thus, together with the results from Experiment 1 (also called Test A1 phase), we tested chimpanzees in the ABA design. The Test A2 phase started from stimulus level L3, which these two chimpanzees reached during the Test A1 phase. The other procedures were identical with Experiment 1 (with $\alpha = 0.05/2$). We compared each chimpanzee's correct responses in target trials across Test A1 and Test B in GLMM and found a significant interaction effect between stimulus species and phase in both chimpanzees (Natsuki: $\chi^2 = 34.61$, df = 1, p<10$^{-8}$; Hatsuka; $\chi^2 = 8.39$, df = 1, p=0.004; also see *Appendix 1—table 2*). We then compared each chimpanzee's performance across Test B and Test A2 and found a significant interaction effect between the two factors in both chimpanzees (Natsuki: $\chi^2 = 37.04$, df = 1, p<10$^{-8}$; Hatsuka; $\chi^2 = 33.75$, df = 1, p<10$^{-8}$). To examine these observed interaction effects further, we performed simple effects tests (*Figure 3* and *Appendix 1—table 2*). Critically, we found that both chimpanzees' performance significantly increased from Test A1 (normal contrast polarity) to Test B (reversed contrast polarity) and then their performance decreased from Test B to Test A2 (normal contrast polarity) in trials presenting the chimpanzee stimuli. Natsuki's performance significantly decreased from Test A1 to Test B and then increased from Test B to Test A2 in trials presenting the human stimuli. Hatsuka's performance did not significantly decrease from Test A1 to Test B but significantly increased from Test B to Test A2 in those trials.

## Discussion

Overall, these results revealed a striking advantage of eyes having positive contrast polarity, namely, the eyes of both species with the uniformly white sclera and a darker iris, in the gaze-discrimination performance of both human and chimpanzee participants. Our results thus supported H1 (perceptual advantage of uniformly white sclera). We also found that, although both human and chimpanzee eye-gaze directions are reliably discernible when those eyes were presented sufficiently large and bright (i.e., L1–2 stimuli), the human eye-gaze directions were more discernible than the chimpanzee eye-gaze particularly when those eyes were presented smaller and more shaded (i.e., L3–4 stimuli).

Our alternative hypotheses (H2–5) cannot explain the overall patterns of our results. H2 (perceptual advantage of the iris-sclera color difference) cannot explain our results likely because it supposes clear visibility of only iris but not that of eye-outline edges, another critical feature that contributes to the

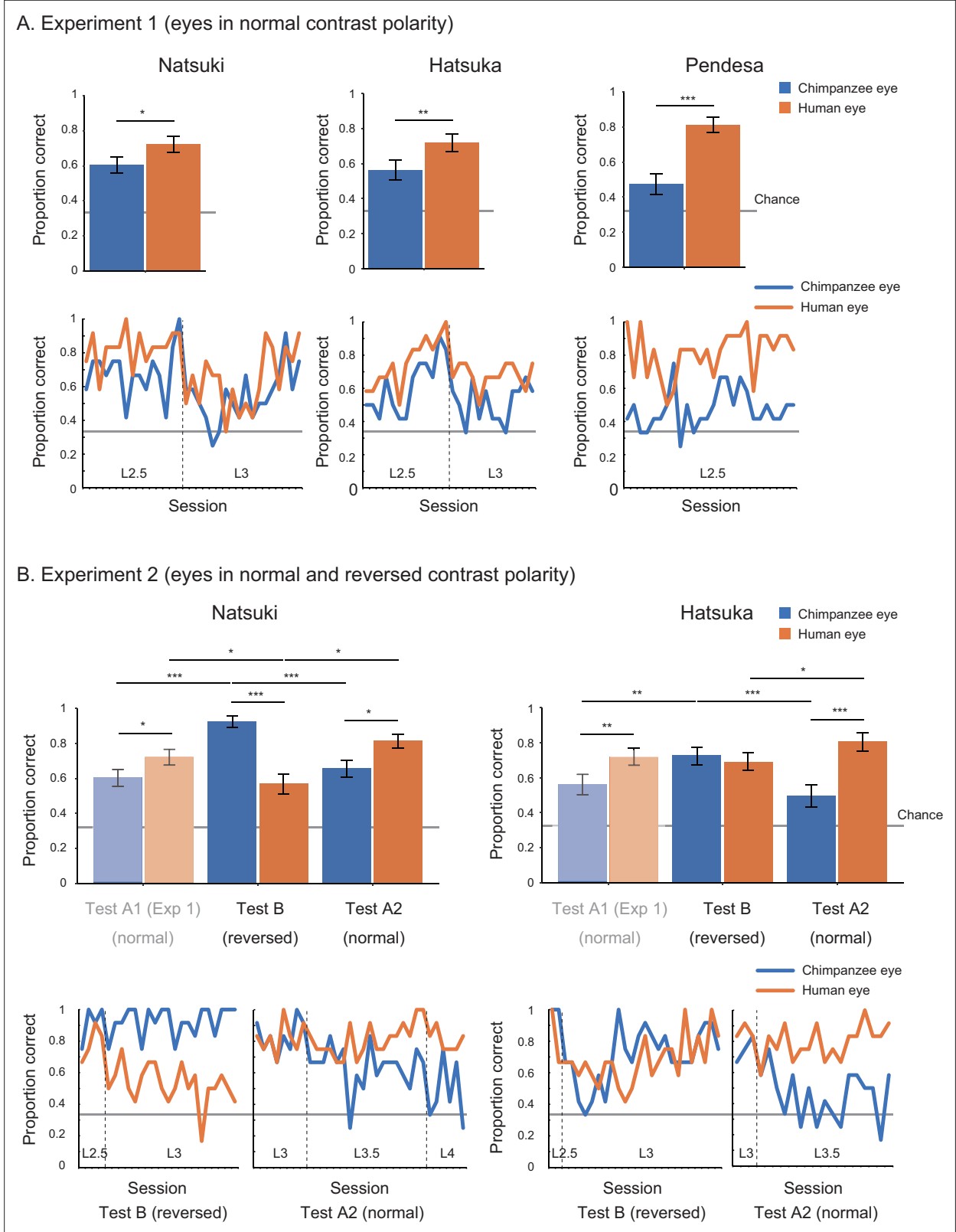

**Figure 3.** Performance of chimpanzee participants (Natsuki, Hatsuka, and Pendesa) in Study 2, represented as mean proportion correct calculated from all sessions (bar graphs) and each session (line graphs). Participants were tested in the ABA design. Specifically, Experiment 1 (**A**) presented eye images with normal contrast polarity (also termed Test A1), and Experiment 2 (**B**) presented eye images with reversed contrast polarity (Test B) and then those having normal contrast polarity (Test A2). The bar graphs of Experiment 2 repeat the same bar graphs of Experiment 1 (with their colors toned

*Figure 3 continued on next page*

*Figure 3 continued*

down) to aid comparisons in the ABA design. Dotted lines in line graphs denote the increment of stimulus level. Recall that the stimulus level varied between L1 and 4 in our stimuli, with L1 being the brightest and largest and L4 being the most shaded and smallest. Error bars in bar graphs are 95% confidence intervals based on a nonparametric bootstrap. Asterisks indicate significance in Experiment 1 and the post-hoc models ran for Experiment 2 (***p<0.001, **p<0.01, *p<0.05; these significance levels were corrected for the number of individuals in each experiment).

visibility of eye-gaze (*Kobayashi and Kohshima, 2001*; *Kano et al., 2021*) (also see *Appendix 3— figure 1*). H3 (perceptual advantage of the horizontally elongated eye shape) also cannot explain our results likely because small variations in the horizontal eye length do not critically affect the visibility of eye-gaze. However, it should be noted that humans can take more extreme sideway eye positions than chimpanzees due to their horizontally elongated eye shape (*Kobayashi and Kohshima, 2001*), and such mechanistic difference may be one advantage of the human eye in eye-gaze signaling. Yet, this fact might be of limited relevance to real-life social interaction because previous eye-tracking studies measuring eye movements of chimpanzees and humans in naturalistic conditions indicated that the majority of eye positions fall within 20°, the eye position adopted in our stimuli, in both species (*Kano and Tomonaga, 2013*; *Kothari et al., 2020*). H4 and H5 (perceptual expertise in own-species and normal eye contrast polarity) also cannot largely explain our results. Finally, one pattern of our results could not be explained by H1 alone, specifically that the negative human eyes and positive chimpanzee eyes affected participants' task performance similarly in some conditions. More specifically, human participants (collectively) performed similarly in trials presenting the human and chimpanzee stimuli in the L4-reversed condition (i.e., the negative human eye and positive chimpanzee eye in the darkest and smallest images). Also, chimpanzee Hatsuka performed similarly in trials presenting the human and chimpanzee stimuli during the Test B (reversed contrast polarity) phase (unlike Natsuki). These partial results are explained by either H2 or the combination of H1 and H3. Overall, however, our results indicate that the presence of the uniformly white sclera (and a darker iris) in our stimuli is the primary factor affecting the similarity of our results between the participants of both species in our study design.

Our results thus suggest that the visibility of human eye-gaze is primarily supported by basic color properties of the eyes, and thus by a basic perceptual mechanism shared among human and chimpanzee participants. Relatedly, there are a number of previous studies documenting the similarities in visual perception, including spectrum sensitivity, visual acuity, and contrast sensitivity, between humans and nonhuman apes (*Deeb et al., 1994*; *Dulai et al., 1994*; *Jacobs et al., 1996*; *Matsuno et al., 2004*; *Matsuno and Tomonaga, 2006*; *Matsuzawa, 1990*; *Bard et al., 1995*; *Adams et al., 2017*) though small differences might exist (*Jacobs et al., 1996*; *Adams et al., 2017*). One notable aspect of our results is that chimpanzees required extensive training to distinguish between different gaze directions of human and chimpanzee eyes, and many of our chimpanzees were unable to pass all the required training phases. Given that many of our chimpanzees were already trained for simple color and form perception tasks (e.g., *Matsuno et al., 2004*; *Matsuzawa, 1990*), this difficulty might suggest that chimpanzees may generally find it more difficult attending to detailed perceptual features of the eyes compared to humans, consistent with the previous training (*Tomonaga and Imura, 2010*) and gaze-following/cueing studies (*Tomasello et al., 2007*; *Tomonaga, 2007*; but see *Povinelli and Eddy, 1996*). In this sense, consistent with the previous experimental studies in humans (*Tomasello et al., 2007*; *Whalen et al., 2004*; *Provine et al., 2013b*; *Ricciardelli et al., 2000*), our results also suggest that gaze perception in humans is supported by both the uniformly white sclera of eyes and their perceptual expertise in such detailed eye features.

One clear limitation of this study is that only a small number of chimpanzees could participate in our test conditions, which hampers the generalization of our results. Nonetheless, if the performance of the current task is supported by simple color properties of eyes and basic visual perception of great apes as argued above, our results should be replicated in other (trained) individuals and likely also in other great ape species. However, further tests are necessary to confirm this prediction. Finally, it remains unanswered whether our results are generalizable to other stimuli with some variations in sclera color, specifically given that some chimpanzee individuals have partly unpigmented (white) sclera (*Perea-García et al., 2019*; *Caspar et al., 2021*; *Mearing and Koops, 2021*; *Kano et al., 2021*). However, such sclera is typically characterized as more graded or patchy compared to humans' uniformly white sclera. Moreover, as noted earlier, the previous simulation study demonstrated that

the visibility of eye-gaze (especially that of eye-outline edges) is limited with the eyes of all nonhuman ape species without the human-like uniformly white sclera, particularly in visually challenging conditions (*Kano et al., 2021*). We thus expect a similar pattern of results even when using eyes with partly unpigmented sclera in our experiments. Again, however, further experimental studies are necessary to confirm this prediction.

In conclusion, we demonstrated that uniform whiteness in the exposed sclera enhances eye-gaze signaling. We thus provided experimental support for the gaze-signaling hypothesis despite recent criticisms on this hypothesis (*Perea-García et al., 2019*; *Caspar et al., 2021*; *Mearing and Koops, 2021*). However, we also propose several significant updates on this hypothesis. Specifically, we found that it is the uniformly white sclera but not necessarily other distinguishing features, such as iris-sclera color difference and some variation in horizontal eye elongation, that critically distinguishes the human eye from the chimpanzee eye in terms of the visibility of eye-gaze direction. Moreover, we found that one function of the uniformly white sclera is to equip the eye-gaze signal with robustness against degradation caused by natural noises (e.g., shading, distancing). These new findings, when combined with the original and related hypotheses, suggest that humans have evolved special external eye morphology for conspecific communication and that it is a vital part of communication for humans to read conspecific eye-gaze cues during their everyday cooperative and cultural activities.

## Materials and methods

### Participants

Study 1 tested 25 human adults (14 females, 11 males; 23 East/South Asians and 1 Caucasian male) who had moderate to extensive experience in caretaking or studying chimpanzees (3 months = 1; 1–5 years = 10; 5–10 years = 4; >10 years = 10). Although our human participants were mostly from similar cultural backgrounds, two related experimental studies tested participants from other cultural backgrounds (*Ricciardelli et al., 2000*; *Yorzinski and Miller, 2020*), and thus our results are complementary to those previous results. Our participants included 10 individuals who had extensive experience interacting with chimpanzees over a decade. We confirmed the same results when we restricted our analyses to those participants. All human participants were workers or students at Kumamoto Sanctuary (KS) or Primate Research Institute (PRI) who were directly invited to participate in this experiment. All were naïve to the experimental hypotheses in this study. All reported having normal to corrected-to-normal vision and no color blindness. Written informed consent was obtained from all participants before the study. The experimental protocol was approved by the internal ethical committee for human experiments in PRI (no. 2020-05).

Study 2 trained 10 chimpanzees (nine females, one male). Among them, three chimpanzees (Natsuki, Hatsuka, Pendesa) passed all the training stages and participated in Experiment 1. Two of these three chimpanzees (Natsuki, Hatsuka) participated in Experiment 2. Daily veterinary checks indicated no specific visual problems (including color blindness) that may have interfered with the execution of current experiments in our chimpanzee participants (though some minor visual problems may exist in Pendesa; *Kaneko et al., 2013*). Chimpanzees lived in a social group of conspecifics at KS or PRI. All chimpanzees were tested in a dedicated testing room at each facility, and their daily participation was voluntary, in that they could decide whether to enter the testing room on a given testing day. They received regular feedings, daily enrichment, and had ad libitum access to water. Animal husbandry complied with institutional guidelines (KS: Wildlife Research Center 'Guide for the Animal Research Ethics'; PRI: 2002 version of 'The Guidelines for the Care and Use of Laboratory Primates'), and the research protocol was approved by the institutional research committee (KS: WRC-2020-KS008A/009A; PRI: 2020-193/209). See *Appendix 2—table 1* for details about participants.

### Apparatus

Study 1 tested human participants in a standard office setting either in KS or PRI. Two participants were tested remotely online given the COVID-19 situation at the time of the experiment. They received the same task program online and performed the task on their computer in a standard office. Although slight differences existed in experimental setups between these two and the other participants (detailed below), we confirmed that including or not including them in our analysis yielded the same results. Participants sat in front of a 23-inch monitor (52.7 × 29.6 cm, SE2416H, Dell, Round

Rock, TX; for one online participant, 52.2 × 29.3 cm, 243V5QHABA/11, Phillips, Amsterdam, the Netherlands; for the other online participant, 50.9 × 28.6 cm xub2390hs-b3, Iiyama, Japan; all monitors were in 1920 × 1080 pixels and set at 100% brightness and 50% contrast). They placed their second- to fourth-digit fingers on the left, down, and right keys of a standard keyboard connected to the computer. With this setup, the viewing distance was about 60–70 cm. Participants were told to sit in front of the monitor as they normally would and not to move their original head position throughout the experiments.

Study 2 tested chimpanzee participants in a testing room equipped with touch panels (ET1790L-7CWB-1-ST-NPB-G, Touch Panel Systems, Yokohama, Japan, in KS; LCD-AD172F2-T, IO-DATA, Kanazawa, Japan, in PRI; both 34.5 × 26.0 cm; both 1280 × 1024 pixels; both 100% brightness and 50% contrast) installed with their centers 45 cm from the floor. With this position, the eye level of the chimpanzees was roughly at the center of the monitor when they sat on the floor. The monitor was installed 15 cm behind transparent polycarbonate panels, and there was a rectangle hole sized 40 × 15 cm on the panel so that chimpanzees could view the stimuli through the panel while making a touch response by inserting their arm through the hole (*Figure 1*). With this setup, the viewing distance was about 30–40 cm. This visual distance for chimpanzee participants is shorter than that for human participants, and thus overall task difficulty should be lower for them, consistent with other procedural differences that we made to ease the task difficulty for chimpanzees.

## Stimuli

We prepared chimpanzee and human facial images with different levels of sizes and brightness. Eyeball regions of these facial images were manipulated to have either normal or reversed contrast polarity (*Appendix 3—figure 3*). To create the chimpanzee stimuli, we selected 10 high-resolution facial images of chimpanzees of both sexes from image collections obtained from colleagues at KS. As we sampled images from KS chimpanzees, some of our stimulus chimpanzees were familiar to KS chimpanzees (half of the stimulus chimpanzees used in both training and test sessions), while all stimulus chimpanzees were unfamiliar to PRI chimpanzees. Yet, we confirmed that KS and PRI chimpanzees did not systematically differ in their performance during the training sessions (see *Appendix 4—figure 1*). All stimulus chimpanzees were familiar to KS human participants, while most stimulus chimpanzees were unfamiliar to PPI human participants. Yet, we confirmed that KS and PRI human participants performed similarly in trials presenting the chimpanzee stimuli.

The selection criteria of stimulus photographs were as follows: (1) the photographed individual oriented both head and eyes directly to the camera, (2) all eye features of the individual were clearly visible, (3) no strong shade was visible on the face, and (4) no expression was shown in the face. See the author's online repository for the full set of chimpanzee stimuli (https://osf.io/2xny3/). The selected chimpanzee individuals included juveniles and adults of both sexes, and their sclera colors were uniformly dark (from the iris edge to the eye corner). Although some chimpanzee individuals have unpigmented spots in their sclera (*Perea-García et al., 2019*; *Caspar et al., 2021*; *Mearing and Koops, 2021*; *Kano et al., 2021*), we selected individuals having relatively uniformly dark sclera to simplify our experimental comparisons (and manipulated them to have dark sclera in eye corners of averted gaze faces; see below for details). To create the human stimuli, we selected 10 high-resolution facial images of humans from the image collections published for research use (*Egger et al., 2011*) based on the same criteria as above. The selected human individuals were teenagers of both sexes in various ethnicities with various skin and eye colors, and their sclera colors were uniformly white (from the iris edge to the eye corner). All stimulus humans were unfamiliar to both chimpanzee and human participants. We balanced the selection of human stimulus individuals so that we could include a wide variety of skin and eye colors among them. Study 1 used the whole sets of chimpanzee and human stimuli, which consisted of 10 stimulus individuals in each stimulus species, and Study 2 used six stimulus individuals in each stimulus species (due to procedural differences between studies; see below).

The facial images of both chimpanzees and humans were then cropped to include only the face and hair and auto-level adjusted to reduce the variations in overall brightness and contrast across images using Photoshop (Adobe, San Jose, CA). The cropped images were then pasted into a uniform 50% gray background sized 400 × 400 pixels. The size of each cropped facial image (for both chimpanzee and human image) was adjusted based on its iris diameter, which was set at 16 pixels (4.2–4.4 mm on the monitors used in both studies 1 and 2). This size adjustment was performed to test the effect

of eye shape (horizontal elongation of eye-opening, related to H3) independently from its absolute size and also to test the effect of white sclera independently from its exposed area size (see below for the quantification of these parameters). It should be noted that, due to these controls, our chimpanzee stimuli were presented as slightly larger than the size proportional to human stimuli because the eyeball size of humans is generally slightly larger (about 5–10%) in that of chimpanzees (*Ross and Kirk, 2007*; *Bekerman et al., 2014*; *Kirk, 2004*). To create facial images with averted gaze, the eyeball part of each face (with direct gaze) was cropped and then shifted six pixels to the side (*Appendix 3—figure 3*; this corresponded to the rotation of the eyeball of about 20° in both stimulus species). We then filled the blank areas in the shifted eye by copying the sclera colors of the original image using the 'stamp' tool in Photoshop. To create the facial images in which the eyes had reversed contrast polarity, we first cropped the eyeball part of each face (with both direct and averted gaze) and then inverted the lightness (or grayscale) component of the cropped part while keeping its chromaticity component unchanged (to avoid unnatural bluish appearance in the eyes) in a custom-made MATLAB program (MathWorks, Natick, MA).

We then evaluated the shape and color of the eyes in our images following a previous method (*Kano et al., 2021*). We first created the region-of-interest (ROI) mask respectively for iris and sclera by tracing and filling the edge of each feature in Photoshop and a custom-made MATLAB program. We then calculated the color of each ROI as the mean of CIELAB color in all pixels within that ROI. CIELAB color system is created to simulate a perceptually uniform color space in humans and is also considered applicable to nonhuman primates with human-like trichromatic color vision (*Stevens et al., 2009*). We then calculated the color difference between iris and sclera in each image as a Euclidean difference between the mean values of these ROIs. We confirmed that, as found in a previous study (*Kano et al., 2021*), the iris-sclera color differences did not significantly differ between our human and chimpanzee stimuli (with both normal and reversed contrast polarities; *Appendix 3—figure 2*). We also measured the shape of the eyes in each image using the same ROIs. We confirmed that, as found in the previous study (*Kano et al., 2021*), the human eye was horizontally longer than the chimpanzee eye, but the size of the sclera ROI did not differ between the human and chimpanzee eye in our stimuli (*Appendix 3—figure 2*).

Finally, we converted the facial images to various levels of sizes and brightness. Study 1 used four stimulus levels (L1–4). L1 stimuli measured 400 pixels in width (original) and 100% brightness (original), L2 stimuli measured 200 pixels in width (1/2) and 50% brightness (1/2), L3 stimuli measured 100 pixels in width (1/4) and 33% brightness (1/3), and L4 stimuli measured 50 pixels in width (1/8) and 25% brightness (1/4). These size and brightness levels were determined based on pilot experiments with two human participants (who did not participate in Study 1) so that the gaze direction of L4 stimuli was recognizable to both participants at least in one of the stimulus species with either positive or negative eyes. In Study 2, we prepared three additional stimulus levels, L1.5, L2.5, and L3.5, which were the intermediate between L1 and 2, L2 and 3, and L3 and 4, respectively, in terms of size and brightness (i.e., L1.5: 300 pixels in width, 75% brightness; L2.5: 150 pixels in width, 42% brightness; L3.5: 75 pixels in width, 29% brightness) so that chimpanzees could move to the next stimulus level without showing substantial drops in their performances (see details about the test procedures below). Studies 1 and 2 used identical stimuli except that Study 1 presented the whole face in a 1:1 square image (e.g., 400 × 400 pixels), while Study 2 presented only the eye region in a 4:1 rectangle image (e.g., 400 × 100 pixels) to reduce attentional demands on chimpanzees.

## Task procedures

We made the task procedures of studies 1 and 2 as similar as possible, although several unavoidable differences existed because chimpanzees required extensive training to master the gaze-detection task. In Study 1, the task for the human participants was to indicate the direction of gaze (left/front/right) in the stimulus face presented at the center of the screen by keypress in each trial. They were instructed to answer as accurately and quickly as possible.

Study 1 consisted of two experiments. All participants completed experiments 1 and 2 in this order. Experiment 1 presented stimuli with normal eye contrast polarity at L1–4 levels. Experiment 2 presented stimuli with eyes having both normal and reversed contrast polarities at L3–4 levels. Before each experiment, they completed 20 practice trials presenting L1 stimuli (with the stimuli with normal eye contrast polarity for Experiment 1 and those with both normal and reversed eye contrast polarities

for Experiment 2). Each experiment consisted of a total of 96 trials with eight blocks (12 trials within each block). In both experiments, each block presented the stimuli of the same species, and the eight blocks alternately presented chimpanzee and human stimuli. In Experiment 1, each block started with three consecutive trials presenting L1 stimuli of either species (with normal eye contrast polarity) and then increased the stimulus level every three trials; namely, 1st–3rd trials, 4th–6th trials, 7th–9th trials, and 10th–12th trials respectively presented L1, L2, L3, and L4 stimuli of either species. Thus, in Experiment 1, 12 trials presented stimuli of either species (chimpanzee, human) at each stimulus level (L1–4). In Experiment 2, each block (12 trials) started with six consecutive trials presenting L3 stimuli of either species and then six consecutive trials presenting L4 stimuli of the same species. The first two blocks (first and second blocks) presented stimuli of the two species with normal eye contrast polarity (each block presented one species), and then the next two blocks (third and fourth blocks) presented stimuli of the two species with reversed eye contrast polarity. The fifth and sixth blocks and the seventh and eighth blocks again presented stimuli of the two species with normal eye contrast polarity and then those with reversed eye contrast polarity. Thus, in Experiment 2, 12 trials presented stimuli of either species (chimpanzee, human) at each stimulus level (L3, L4) in either eye contrast polarity (normal, reversed). Each participant completed all experiments in 25–30 min. All experiments were conducted in November 2020.

The number of times in which each gaze direction (left/front/right) was presented was balanced in each participant (i.e., each direction was presented in 32 trials per participant), and the number of times in which each stimulus individual was presented was also balanced both within each participant and across participants (i.e., each stimulus individual was presented on average 4.8 times per participant). The order of presenting the chimpanzee or human stimuli in the first block was counterbalanced across participants. The orders of gaze directions (left/front/right) and stimulus individuals were pseudorandomized so that the same gaze direction was not presented in more than two successive trials, and the same stimulus individual was not presented in any successive trials.

In Study 2, the task for the chimpanzee participants was to indicate the image with averted gaze (shifted to the right; called the target image) among two other images with direct gaze (called the distractor images) by a touch response in each trial (i.e., three-item visual search task, following the task design by *Tomonaga and Imura, 2010*). The target and distractor images differed only in their eye-gaze direction (but not in their eye contrast polarity or brightness/size). The three images were centered at 220, 640 (center), and 860 pixels horizontally and 512 pixels vertically on a 1280 × 1024 pixels monitor. Chimpanzees were given a sip of grape juice or a piece of apple (depending on their preference) when they answered correctly in each trial (the same amount of reward was given for each chimpanzee throughout the study). Before training, we performed a pilot experiment (200–600 trials for each chimpanzee) to decide the general task design, especially in terms of the number of distractors in each trial, the number of trials in each session, and the features of initial stimuli, so that the chimpanzees could gradually learn the task.

As in Study 1, Study 2 consisted of two experiments. Experiment 1 presented stimuli with normal eye contrast polarity, and Experiment 2 presented those with reversed eye contrast polarity and then those with normal eye contrast polarity. Thus, these two experiments presented stimuli with normal and reversed eye contrast polarities in the ABA design. Throughout Study 2 (both training and test), each session consisted of 48 trials and four blocks. Each block (12 trials) presented stimuli of the same species, and the four blocks alternately presented the chimpanzee and human stimuli. Each chimpanzee performed 1–8 sessions per day depending on their motivation. Each session lasted about 10 min. Study 2 took about 8 months from August 2020 to March 2021 including both training and test periods.

Training performed before these two experiments in Study 2 consisted of six training stages, and chimpanzees were trained for the task in a step-by-step manner through these stages. Training stage 1 presented the target image with no iris with the distractor images with irises (direct gaze; see *Appendix 3—figure 4* for the examples). Training stages 2–4 presented the target image in which the iris was positioned in 38°, 30°, and then 20° (the final iris position was 20°; *Appendix 3—figure 4*), following the training procedure employed by a previous study (*Tomonaga and Imura, 2010*). Training stages 1–4 used two stimulus individuals per stimulus species, and training stages 5 and 6 added two new stimulus individuals per stimulus species in each stage; thus, training stages 5 and 6 respectively presented four and six stimulus individuals per stimulus species (the final stimulus set

in training stage 6). The criterion of passing each training stage was either scoring over 90% in one session or 80% in two consecutive sessions both in trials presenting the chimpanzee stimuli and those presenting the human stimuli. We trained chimpanzees in the same number of trials for the human and chimpanzee stimuli to avoid biasing their learning for either stimulus species. Eight chimpanzees learned this visual search task with the most basic stimulus set (training stage 1) and continued to the next training stages (training stages 2–6). During the latter training stages, there was no consistent bias across individuals in their performances for the trials presenting the human and chimpanzee stimuli. However, some chimpanzees performed notably better in trials presenting the chimpanzee stimuli than those presenting the human stimuli (Cleo, Iroha, Mizuki) and some showed the opposite pattern (Pendesa), while others performed similarly in those trials (*Appendix 4—figure 1*). These observed individual differences do not appear to be related to the particular backgrounds of each individual (*Appendix 1—table 1*). Three chimpanzees (Natsuki, Hatsuka, Pendesa) passed all training stages after extensive training (*Appendix 4—table 2*) and learned to reliably differentiate the eye-gaze directions of both humans and chimpanzee stimuli (L1, normal eye contrast polarity; *Appendix 4—figure 1*). Three chimpanzees passed all the training stages, namely, that they performed reliably in both trials presenting the chimpanzee and human stimuli and then participated in Experiment 1. See *Appendix 4—table 2* for the number of sessions each chimpanzee had in each training stage.

Experiment 1 was divided into pre-test and test phases (pre-Test A1 and Test A1 phases). To test chimpanzees at stimulus levels higher than L1, we incremented the stimulus level by 0.5 when the chimpanzee scored above 85% in two successive sessions (in all trials at stimulus level L1 and test trials at stimulus level higher than L1; see below for details about the test and baseline trials). The pre-test phase started from stimulus level L1 and the test phase started from stimulus level L2.5. We used stimulus levels higher than (or equal to) L2.5 for the test phase because the results from Study 1 (human participants) indicated that clear performance differences between the stimulus species (i.e., test conditions) emerged at stimulus levels higher than L2. To examine individuals' performance across sessions, each individual completed a minimum of 20 test sessions. To avoid the ceiling effect, we incremented the stimulus level by 0.5 when the individual scored above 85% in two successive sessions in test trials during the test phase. Two chimpanzees (Natsuki and Hatsuka) participated in Experiment 2. The other chimpanzee (Pendesa) took more than twice as many sessions as the other two chimpanzees for training and thus was dropped from Experiment 2 (*Appendix 4—table 2* and *Figure 1*). Experiment 2 first presented stimuli with reversed eye contrast polarity (pre-Test B and Test B phases) and then presented those with normal eye contrast polarity (pre-Test A2 and Test A2 phases). As in Experiment 1, the pre-Test B and pre-Test A2 presented L1–2 stimuli, and the Test B and Test A2 phase presented L2.5–4 stimuli. Each chimpanzee completed a minimum of 20 test sessions respectively in the Test B and Test A2 phases. The other procedures were identical to those in Experiment 1. It should be noted that, although the number of these test sessions varied across individuals and test phases, we confirmed that limiting the dataset to 20 sessions in all individuals and sessions yielded the same results. See *Appendix 4—table 3* for the number of sessions each chimpanzee had in each pre-test and test stage.

In both experiments 1 and 2, (pre-)test sessions presenting stimulus levels higher than or equal to L1.5 consisted of 24 baseline and 24 test trials. The baseline trials presented L1 stimuli, and the test trials presented the stimuli at higher levels. Each block (12 trials) presented six baseline trials consecutively and then six test trials. Thus, in each test session, 12 trials presented stimuli of either species (chimpanzee, human) at the L1 (baseline trials) or higher levels (test trials). Chimpanzees maintained high performances in baseline trials across sessions for both human and chimpanzee stimuli in the Test A1 phase (L1 stimuli with normal color; Natsuki: 89% ± 12% vs. 90% ± 8%; Hatsuka: 91% ± 10% vs. 89% ± 8%; Pendesa; 97% ± 5% vs. 91% ± 7%; mean ± SD), Test B phase (L1 stimuli with reversed eye contrast polarity; Natsuki: 91% ± 7% vs. 94% ± 7%; Hatsuka: 92% ± 10% vs. 96% ± 6%; mean ± SD), and Test A2 phase (L1 stimuli with normal eye contrast polarity; Natsuki: 91% ± 10% vs. 94% ± 7%; Hatsuka: 92% ± 7% vs. 96% ± 5; mean ± SD).

As in Study 1, the number of times in which the target image (with averted gaze) was presented on each location (left/center/right) was balanced in each session (i.e., each gaze direction was presented in 16 trials per session), and the number of times in which each stimulus individual was presented was balanced in each session (i.e., each stimulus individual was presented four times per session). The order of presenting chimpanzee or human images in the first block was counterbalanced across

sessions. The locations of the target images and the order of presenting the stimulus individuals were pseudorandomized so that the target image did not appear on the same location in more than two successive trials, and the same stimulus individual was not presented in any successive trials. See *Appendix 4—table 1* for the summarized descriptions of each stage at the training and (pre-)test phases and *Appendix 4—table 2* and *Appendix 4—table 3* for the number of sessions in each stage at the training and (pre-)test phases, respectively.

## Data analysis

To test the participants' performance differences between conditions, we ran a binomial GLMM in R (version 4.0.5). In Experiment 1 of Study 1, the model included participants' correct response (correct, incorrect) as the response variable, stimulus species (chimpanzee image, human image), and stimulus level (L1–4) as test fixed factors, the interaction between those test factors, block and (within-block) trial, which was nested in each block, as control fixed factors, and participant and stimulus individual as random factors (see *Appendix 1—table 1* for the formulas). In Experiment 2 of Study 1, we used the same model with eye contrast polarity (normal, reversed) as an additional test fixed factor (and its interaction with the other test factors; *Appendix 1—table 1*). In Study 2 (chimpanzees), as we evaluated chimpanzees' performance in repeated sessions and adjusted their performances by incrementing stimulus levels according to their performance, we treated the session as a random factor and did not include stimulus level in the model. Therefore, the model included stimulus species as a test fixed factor, block and (within-block) trial as control fixed factors, and stimulus individual and sequence as random factors (*Appendix 1—table 1*). Study 2 performed statistical tests for each chimpanzee with the $\alpha$ level adjusted for the number of individuals in the Bonferroni correction; namely, 0.05/3 in Experiment 1 and 0.05/2 in Experiment 2. For all models in studies 1 and 2, we included all possible random slope components, although we removed the correlations between random slopes and intercepts to ease the nonconvergence issues (*Barr et al., 2013*). Overdispersion was checked using the dispersion parameters derived from the R package 'blmeco' and did not seem to be an issue in any of our models (they ranged between 0.77 and 1.12). The significance of a given term was tested using a likelihood ratio test. Nonsignificant interaction terms were dropped to test the significance of lower-order terms. When the interaction term was significant, post-hoc comparisons were performed to examine simple effects at each factor level. All data and R scripts are available in our online repository (https://osf.io/2xny3/).

## Acknowledgements

We thank our colleagues at Kumamoto Sanctuary and Primate Research Institute (especially, Etsuko Ichino, Ayumu Santa, Akiho Muramatsu, and André Gonçalves and Drs. Naruki Morimura and Ikuma Adachi) for their assistance in performing experiments. We also thank Drs. Christopher Krupenye and Lydia Hopper for their edits and fruitful comments. Financial support came from the Japan Society for the Promotion of Science KAKENHI Grants 19H01772 and 20H05000 to FK and 18J20077 to YK.

## Additional information

### Funding

| Funder | Grant reference number | Author |
| --- | --- | --- |
| Japan Society for the Promotion of Science | 19H01772 | Fumihiro Kano |
| Japan Society for the Promotion of Science | 20H05000 | Fumihiro Kano |
| Japan Society for the Promotion of Science | 18J20077 | Yuri Kawaguchi |

The funders had no role in study design, data collection and interpretation, or the decision to submit the work for publication.

### Author contributions
Fumihiro Kano, Conceptualization, Formal analysis, Funding acquisition, Investigation, Methodology, Writing – original draft, Writing – review and editing; Yuri Kawaguchi, Yeow Hanling, Investigation, Writing – review and editing

### Author ORCIDs
Fumihiro Kano (iD) http://orcid.org/0000-0003-4534-6630

### Ethics
Human subjects: Written informed consent was obtained from all participants prior to the study. Experimental protocol was approved by the internal ethical committee for human experiments in Primate Research Institute, Kyoto University (No. 2020-05).
Animal husbandry complied with institutional guidelines (KS: Wildlife Research Center "Guide for the Animal Research Ethics"; PRI: 2002 version of "The Guidelines for the Care and Use of Laboratory Primates"), and research protocol was approved by the institutional research committee (KS: WRC-2020-KS008A/009A; PRI: 2020-193/209).

### Decision letter and Author response
Decision letter https://doi.org/10.7554/eLife.74086.sa1
Author response https://doi.org/10.7554/eLife.74086.sa2

---

## Additional files

### Supplementary files
• Transparent reporting form

### Data availability
All data are available in our online repository (https://osf.io/2xny3/).

The following dataset was generated:

| Author(s) | Year | Dataset title | Dataset URL | Database and Identifier |
|-----------|------|---------------|-------------|-------------------------|
| Kano F | 2021 | Experimental evidence for the gaze signaling hypothesis | https://osf.io/2xny3/ | Open Science Framework, 2xny3 |

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

# Appendix 1

## Result: GLMM

**Appendix 1—table 1.** The R formulas for generalized linear mixed model (GLMM) in studies 1 and 2.

| Test | Participant | R formula |
|---|---|---|
| Study 1: Exp. 1 | Humans | glmer(Correct ~ Species*Level + Block/Trial + (1+ Species*Level + Block/Trial \|\| ParticipantID) + (1+ Level + Block/Trial \|\| StimulusID), family = binomial, data = exp1_humans) |
| Study 1: Exp. 2 | Humans | glmer(Correct ~ Species*Level*Polarity + Block/Trial + (1+ Species*Level*Polarity + Block/Trial \|\| ParticipantID) + (1+ Level*Polarity + Block/Trial \|\| StimulusID), family = binomial, data = exp2_humans) |
| Study 2: Exp. 1 | Chimpanzees | glmer(Correct ~ Species + Block/Trial+ (1+ Block/Trial \|\| StimulusID) + (1+ Block/Trial \|\| SequenceID), family = binomial, data = exp1_eachchimpparticipant) |
| Study 2: Exp. 2 | Chimpanzees | glmer(Correct ~ Species*Phase + Block/Trial + (1+ Phase + Block/Trial \|\| StimulusID) + (1+ Block/Trial \|\| SequenceID), family = binomial, data = exp2_eachchimpparticipant) |

Correct (correct response; correct/incorrect), Species (stimulus species; chimpanzee/human), Level (stimulus level; L1–4), Polarity (contrast polarity; positive/negative), Phase (test phase; A1/B/A2)

**Appendix 1—table 2.** The results from generalized linear mixed model (GLMM) in studies 1 and 2.

| Experiment | Participant | Effect | $\beta$ | SE | $\chi^2$ | d.f. | p | $\alpha^*$ |
|---|---|---|---|---|---|---|---|---|
| Study 1: Exp. 1 | Humans | Level × Species[*†] | 0.29 (L1 vs. L2) | 1.96 (L1 vs. L2) | | | | |
| | | | 1.09 (L1 vs. L3) | 1.88 (L1 vs. L3) | | | | |
| | | | −0.17 (L1 vs. L4) | 1.76 (L1 vs. L4) | 3.47 | 3 | 0.32 | 0.05 |
| | | Level | −1.04 (L1 vs. L2) | 0.52 (L1 vs. L2) | | | | |
| | | | −2.43 (L1 vs. L3) | 0.50 (L1 vs. L3) | | | | |
| | | | −3.76 (L1 vs. L4) | 0.56 (L1 vs. L4) | 45.85 | 3 | $<10^{-9}$ | 0.05 |
| | | Species | 2.83 | 0.35 | 37.08 | 1 | $<10^{-8}$ | 0.05 |
| Study 1: Exp. 2 | Humans | Level × Species × Polarity | 3.35 | 0.88 | 17.57 | 1 | $<10^{-4}$ | 0.05 |
| Post-hoc (L3, normal) | | Species | 3.47 | 0.69 | 40.00 | 1 | $<10^{-9}$ | 0.05 |
| Post-hoc (L3, inverted) | | Species | 3.02 | 0.65 | 22.28 | 1 | $<10^{-5}$ | 0.05 |
| Post-hoc (L4, normal) | | Species | 1.05 | 0.29 | 30.17 | 1 | $<10^{-7}$ | 0.05 |
| Post-hoc (L4, inverted) | | Species | 0.47 | 0.27 | 2.91 | 1 | 0.088 | 0.05 |
| Post-hoc (human, L3) | | Polarity | 3.82 | 0.51 | 30.77 | 1 | $<10^{-7}$ | 0.05 |
| Post-hoc (chimp, L3) | | Polarity | 3.27 | 0.71 | 19.74 | 1 | $<10^{-5}$ | 0.05 |
| Post-hoc (human, L4) | | Polarity | 1.94 | 0.35 | 17.86 | 1 | $<10^{-4}$ | 0.05 |
| Post-hoc (chimp, L4) | | Polarity | 0.92 | 0.25 | 7.89 | 1 | 0.005 | 0.05 |
| Study 2: Exp. 1 (Test-A1) | Natsuki | Species | 0.55 | 0.16 | 8.28 | 1 | 0.004 | 0.05/3 |
| | Hatsuka | Species | 0.71 | 0.19 | 9.50 | 1 | 0.002 | 0.05/3 |
| | Pendesa | Species | 1.69 | 0.24 | 21.94 | 1 | $<10^{-5}$ | 0.05/3 |
| Study 1: Exp. 1–2 (Test A1 vs. B) | Natsuki | Phase × Species | 3.40 | 0.41 | 34.61 | 1 | $<10^{-8}$ | 0.05/2 |
| | Hatsuka | Phase × Species | 1.62 | 0.40 | 8.39 | 1 | 0.004 | 0.05/2 |
| Study 1: Exp. 2 (Test B vs. A2) | Natsuki | Phase × Species | 8.49 | 0.86 | 37.04 | 1 | $<10^{-8}$ | 0.05/2 |
| | Hatsuka | Phase × Species | 3.42 | 0.75 | 33.75 | 1 | $<10^{-8}$ | 0.05/2 |

*Appendix 1—table 2 Continued on next page*

*Appendix 1—table 2 Continued*

| Experiment | Participant | Effect | $\beta$ | SE | $\chi^2$ | d.f. | p | $\alpha^*$ |
|---|---|---|---|---|---|---|---|---|
| Post-hoc (Test B) | Natsuki | Species | 2.30 | 0.48 | 12.83 | 1 | 0.003 | 0.05/2 |
|  | Hatsuka | Species | 0.13 | 0.40 | 0.10 | 1 | 0.75 | 0.05/2 |
| Post-hoc (Test A2) | Natsuki | Species | 0.92 | 0.30 | 7.24 | 1 | 0.007 | 0.05/2 |
|  | Hatsuka | Species | 1.46 | 0.27 | 15.11 | 1 | 0.0001 | 0.05/2 |
| Post-hoc (human, Test A1 vs. B) | Natsuki | Species | 0.68 | 0.24 | 7.03 | 1 | 0.008 | 0.05/2 |
|  | Hatsuka | Species | 0.09 | 0.21 | 0.18 | 1 | 0.67 | 0.05/2 |
| Post-hoc (human, Test B vs. A2) | Natsuki | Species | 1.25 | 0.42 | 5.33 | 1 | 0.021 | 0.05/2 |
|  | Hatsuka | Species | 0.60 | 0.22 | 7.22 | 1 | 0.007 | 0.05/2 |
| Post-hoc (chimp, Test A1 vs. B) | Natsuki | Species | 2.15 | 0.29 | 24.74 | 1 | $<10^{-6}$ | 0.05/2 |
|  | Hatsuka | Species | 0.85 | 0.26 | 9.57 | 1 | 0.002 | 0.05/2 |
| Post-hoc (chimp, Test B vs. A2) | Natsuki | Species | 1.99 | 0.002 | 33.22 | 1 | $<10^{-8}$ | 0.05/2 |
|  | Hatsuka | Species | 1.09 | 0.27 | 14.89 | 1 | 0.0001 | 0.05/2 |

Species (stimulus species; chimpanzee/human), Level (stimulus level; L1-4), Polarity (contrast polarity; positive/negative), Phase (test phase; A1/B/A2).

*$\alpha$ level was adjusted for the number of individuals in Study 2.

†These nonsignificant interaction terms were dropped to test the main effects in these models.

# Appendix 2

## Method: Participant

**Appendix 2—table 1.** Details about the chimpanzee participants.

| Participant | Group | Sex | Age | Rearing condition | Participated in |
|---|---|---|---|---|---|
| Ai | PRI | F | 41 | Nursery/peers** | Training |
| Ayumu | PRI | M | 20 | Mother** | Training |
| Chloe[†] | PRI | F | 40 | Nursery/peers | Training |
| Cleo | PRI | F | 20 | Mother | Training |
| Pal | PRI | F | 20 | Mother | Training |
| Pendesa | PRI | F | 43 | Nursery/peers | Training, experiment 1 |
| Hatsuka | KS | F | 12 | Nursery/peers | Training, experiments 1–2 |
| Iroha | KS | F | 12 | Mother | Training |
| Mizuki | KS | F | 24 | Nursery/peers | Training |
| Natsuki | KS | F | 15 | Mother | Training, experiments 1–2 |

Two additional chimpanzees (Zamba and Misaki) participated in a few pilot sessions but did not participate in the training sessions due to low motivation.

*Nursery/peers indicates that individuals were reared by human caretakers and peer conspecifics, while mother indicates that they were reared by their biological mothers.

[†]Chloe was involved in a related gaze-direction search task in a previous study (Tomonaga and Imura, 2010).

## Appendix 3

### Method: Stimuli

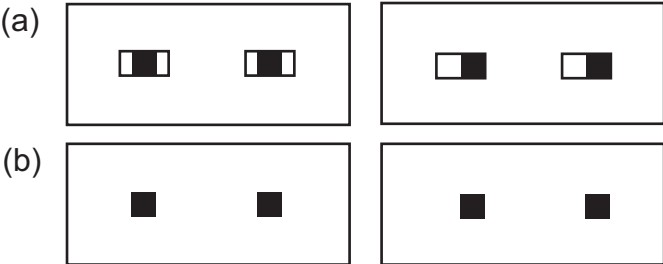

**Appendix 3—figure 1.** Schematic illustration of eyes when both iris and eye-outline edges are visible (**A**) and only iris is visible (**B**). Note that eye-gaze direction is more clearly discernible when both features are visible compared to when only iris is visible (*Kobayashi and Kohshima, 2001*; *Kano et al., 2021*).

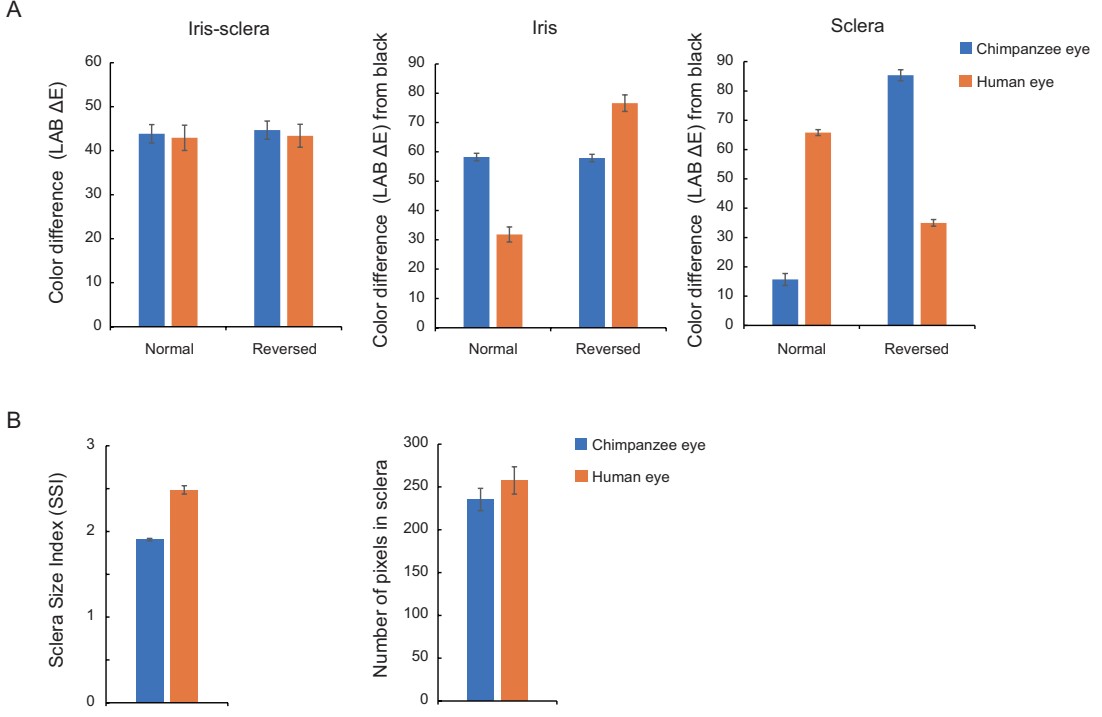

**Appendix 3—figure 2.** Quantification of eye color (**A**) and eye shape (**B**) in the stimuli. Color difference was quantified as the Euclidean distance between the two CIELAB colors $\sqrt{(L1-L2)^2 + (a1-a2)^2 + (b1-b2)^2}$, following a previous study (*Kano et al., 2021*). The colors of the iris and sclera were the means of CIELAB colors of all pixels respectively in the iris and sclera regions of interest (ROI). The iris-sclera color difference was the difference between those two means. The color brightness of the iris and sclera was the difference between the mean of each color and the black (L = 0, a = 0, b = 0). The iris-sclera color difference did not significantly differ between the stimulus species in either normal or inverted color (*t*-test; normal: $t18 = 0.26$, $p=0.80$, $d = 0.12$; inverted: $t18 = 0.39$, $p=0.70$, $d = 0.17$). The eye shape was evaluated using the Sclera Size Index (SSI), calculated as the longest length of eye-opening divided by the iris diameter (Kobayashi and Kohshima, 2001). The human eye was horizontally longer than the chimpanzee eye, as indicated by higher SSI ($t18 = 11.34$, $p<10^{-3}$, $d = 5.07$). We also measured the area size of the sclera in the human and chimpanzee eye as the number of pixels in the sclera ROI. The area size did not significantly differ between the stimulus species ($t18 = 1.08$, $p=0.29$, $d = 0.48$). These statistical comparisons were performed using the full stimulus set used for Study 1 (20 images), but the same results were obtained using the stimulus set used for Study 2 (12 images).

|  | Chimpanzee images | Human images |
|---|---|---|

Direct gaze,
 normal contrast polarity

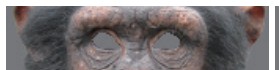 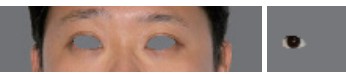

*Cropping out eyeball images*

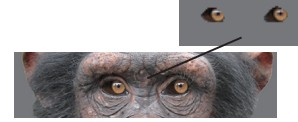 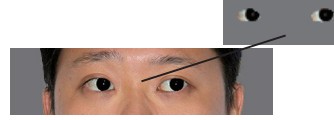

*Shifting the cropped eyeball images to the side*

*and filling the blank parts with the same sclera colors*

Averted gaze,
 normal contrast polarity

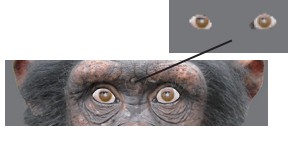 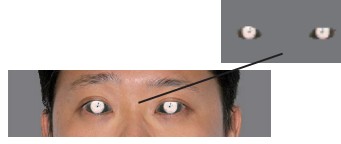

*Reversing the lightness values of the cropped eyeball images (in LAB color)*

Direct gaze,
 reversed contrast polarity

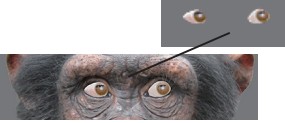 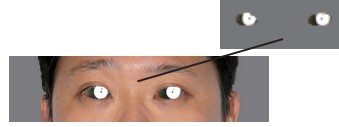

Averted gaze,
 reversed contrast polarity

**Appendix 3—figure 3.** Stimulus preparation. To create the facial images with averted gaze, we first cropped the eyeball part of each face with a direct gaze and then shifted it 6 pixels to the side (this corresponded to the rotation of the eyeball of about 20° in both stimulus species). We then filled the blank areas in the shifted eye by copying the sclera colors of the original image. To create the facial images with reversed contrast polarity, we first cropped the eyeball part of each face (with both direct and averted gaze) and then inverted its lightness component while keeping its chromatic component unchanged. Permission was obtained to publish the human image (this image was only for presentation purposes; not used in this study but edited following the methods used in this study; see *Egger et al., 2011* for the stimuli used in the study).

|  | Human images | Chimpanzee images |
|---|---|---|
| Direct gaze | | |
| No iris | | |
| Eyes shifted 38 degrees | | |
| Eyes shifted 30 degrees | | |
| Eyes shifted 20 degrees (final position) | | |

**Appendix 3—figure 4.** The stimuli used for training chimpanzee participants. Chimpanzees were trained for the task in a step-by-step manner with training stage 1 presenting the target image with no iris with the distractor images with direct gaze, and training stages 2–4 presenting the target image in which the iris was positioned in 38°, 30°, and 20°, following the training procedure employed by a previous study (Tomonaga and Imura, 2010). Permission was obtained to publish the human image (this image was only for presentation purposes; not used in this study but edited following the methods used in this study; see *Egger et al., 2011* for the stimuli used in the study).

# Appendix 4

## Method: Training and pre-test

**Appendix 4—table 1.** Details about each training and test stage for the chimpanzees.

| Training/test phase | Training/test stage | Description | Number of stimulus individual | Stimulus properties | |
|---|---|---|---|---|---|
| | | | | Size (width × height in pixel) | Brightness (0–100% of the original RGB values) |
| | Training 1 | Presenting stimuli in which the iris was removed (i.e., only sclera was visible in the eye). | | | |
| | Training 2 | Presenting stimuli in which eyes were averted 38° (the iris was visible in the corner of the eye). | | | |
| | Training 3 | Presenting stimuli in which the eyes were averted 30°. | | | |
| | Training 4 | Presenting stimuli in which the eyes were averted 20° (the final position of the iris). | 4 (two chimpanzees, two humans) | | |
| | Training 5 | Presenting four new stimulus individuals in half of the trials and four old stimulus individuals in the other half. | 8 (four chimpanzees, four humans) | | |
| Training | Training 6 | | | | |
| | L1 normal | Presenting L1 stimuli (original size and brightness) in each session. Contrast polarities of eyes are normal in all stimuli. | | 400 × 100 | 100 |
| | L1.5 normal | | | 300 × 75 (in test trial) | 75 (in test trial) |
| Pre-Test A1/A2 | L2 normal | | | 200 × 50 (in test trial) | 50 (in test trial) |
| | L2.5 normal | Presenting L1 stimuli in 24 baseline trials and stimuli with a higher level (smaller and darker) in 24 test trials. Contrast polarities of eyes are normal in all stimuli. | | 150 × 37.5 (in test trial) | 42 (in test trial) |
| | L3 normal | | | 100 × 25 (in test trial) | 33 (in test trial) |
| | L3.5 normal | | | 75 × 18.75 (in test trial) | 29 (in test trial) |
| Test A1/A2 | L4 normal | | | 50 × 12.5 (in test trial) | 25 (in test trial) |
| | L1 reversed | | | 400 × 100 | 100 |
| | L1.5 reversed | | | 300 × 75 (in test trial) | 75 (in test trial) |
| Pre-Test B | L2 reversed | | | 200 × 50 (in test trial) | 50 (in test trial) |
| | L2.5 reversed | | | 150 × 37.5 (in test trial) | 42 (in test trial) |
| | L3 reversed | | | 100 × 25 (in test trial) | 33 (in test trial) |
| | L3.5 reversed | Same as L1–4. Normal except that contrast polarities of eyes are reversed in all stimuli. | 12 (six chimpanzees, six humans) | 75 × 18.75 (in test trial) | 29 (in test trial) |
| Test B | L4 reversed | | | 50 × 12.5 (in test trial) | 25 (in test trial) |

See Appendix 3—figure 4 for the training stimuli.

**Appendix 4—table 2.** The number of sessions in each training stage.

| Participant | Stage | Number of sessions |
|---|---|---|
| Ai | Training 1 | *4 |
| | Training 2 | 51 |
| | Training 3 | 15 |
| | Training 4 | 28 |

*Appendix 4—table 2 Continued on next page*

*Appendix 4—table 2 Continued*

| Participant | Stage | Number of sessions |
|---|---|---|
| | Total | 98 |
| Ayumu | Training 1 | 23* |
| | Training 2 | 7 |
| | Total | 30 |
| Chloe | Training 1 | 2 |
| | Training 2 | 42 |
| | Total | 44 |
| Cleo | Training 1 | 9 |
| | Training 2 | 21 |
| | Total | 30 |
| Pal | Training 1 | 26 |
| | Total | 26 |
| Pendesa† | Training 1 | 2 |
| | Training 2 | 29 |
| | Training 3 | 8 |
| | Training 4 | 14 |
| | Training 5 | 11 |
| | Training 6 | 50 |
| | Total | 114 |
| Hatsuka† | Training 1 | 15 |
| | Training 2 | 13 |
| | Training 3 | 7 |
| | Training 4 | 11 |
| | Training 5 | 5 |
| | Training 6 | 3 |
| | Total | 54 |
| Iroha | Training 1 | 33 |
| | Training 2 | 46 |
| | Training 3 | 12 |
| | Total | 91 |
| Mizuki | Training 1 | 14 |
| | Training 2 | 98 |
| | Total | 112 |
| Natsuki† | Training 1 | 4 |
| | Training 2 | 26 |
| | Training 3 | 5 |
| | Training 4 | 15 |
| | Training 5 | 5 |

*Appendix 4—table 2 Continued on next page*

*Appendix 4—table 2 Continued*

| Participant | Stage | Number of sessions |
|---|---|---|
| | Training 6 | 2 |
| | Total | 57 |

*Ai and Ayumu were mistakenly moved to training 2 after only one session scoring >80% in both chimpanzee and human trials (i.e., one additional session was necessary to pass the criteria). For Ai, we performed one additional training 1 session during training 2, confirmed that she scored >80% in both chimpanzee and human trials, and then continued her training. Ayumu performed seven training 2 sessions after training 1, but due to his low motivation to participate in this experiment, we decided to drop him from further tests (we also dropped those seven training 2 sessions from the analysis).
†These three individuals passed all the training stages.

**Appendix 4—table 3.** The number of sessions in each pre-test and test stage.

| Participant | Test phase | Stage | Number of sessions |
|---|---|---|---|
| Pendesa | Pre-Test A1 | L1 normal | 12 |
| | | L1.5 normal | 4 |
| | | L2 normal | 19 |
| | | Total | 35 |
| | Test A1 | L2.5 normal | 26 |
| | | Total | 26 |
| Hatsuka | Pre-Test A1 | L1 normal | 5 |
| | | L1.5 normal | 8 |
| | | L2 normal | 3 |
| | | Total | 16 |
| | Test A1 | L2.5 normal | 13 |
| | | L3 normal | 13 |
| | | Total | 26 |
| | Pre-Test B | L1 reversed | 10 |
| | | L1.5 reversed | 2 |
| | | L2 reversed | 4 |
| | | Total | 16 |
| | Test B | L2.5 reversed | 2 |
| | | L3 reversed | 24 |
| | | Total | 26 |
| | Pre-Test A2 | L2 normal | 1 |
| | | L2.5 normal | 1* |
| | | Total | 2 |
| | Test A2 | L3 normal | 3 |
| | | L3.5 normal | 17 |
| | | Total | 20 |
| Natsuki | Pre-Test A1 | L1 normal | 4 |
| | | L1.5 normal | 5 |

*Appendix 4—table 3 Continued on next page*

*Appendix 4—table 3 Continued*

| Participant | Test phase | Stage | Number of sessions |
|---|---|---|---|
| | | L2 normal | 6 |
| | | L3 normal | 2[†] |
| | | Total | 17 |
| | Test A1 | L2.5 normal | 15 |
| | | L3 normal | 18 |
| | | Total | 33 |
| | Pre-Test B | L1 reversed | 10 |
| | | L1.5 reversed | 10 |
| | | L2 reversed | 3 |
| | | Total | 23 |
| | Test B | L2.5 reversed | 4 |
| | | L3 reversed | 20 |
| | | Total | 24 |
| | Pre-Test A2 | L1.5 normal | 1 |
| | | L2 normal | 1 |
| | | L2.5 normal | 1[*] |
| | | Total | 3 |
| | Test A2 | L3 normal | 8 |
| | | L3.5 normal | 18 |
| | | L4 normal | 6 |
| | | Total | 32 |

*These L2.5 sessions in the Test A2 phase were performed to confirm that the participants' performances did not drop significantly from those in the Test A1 phase. Test A2 phase started from L3, the level that these participants reached in the Test A1 phase.

†As Natsuki performed poorly on these first two L3 sessions, we leveled down the stimuli to L2.5. These initial L3 sessions were not included in the analysis (yet including or not including these two sessions did not change the results).

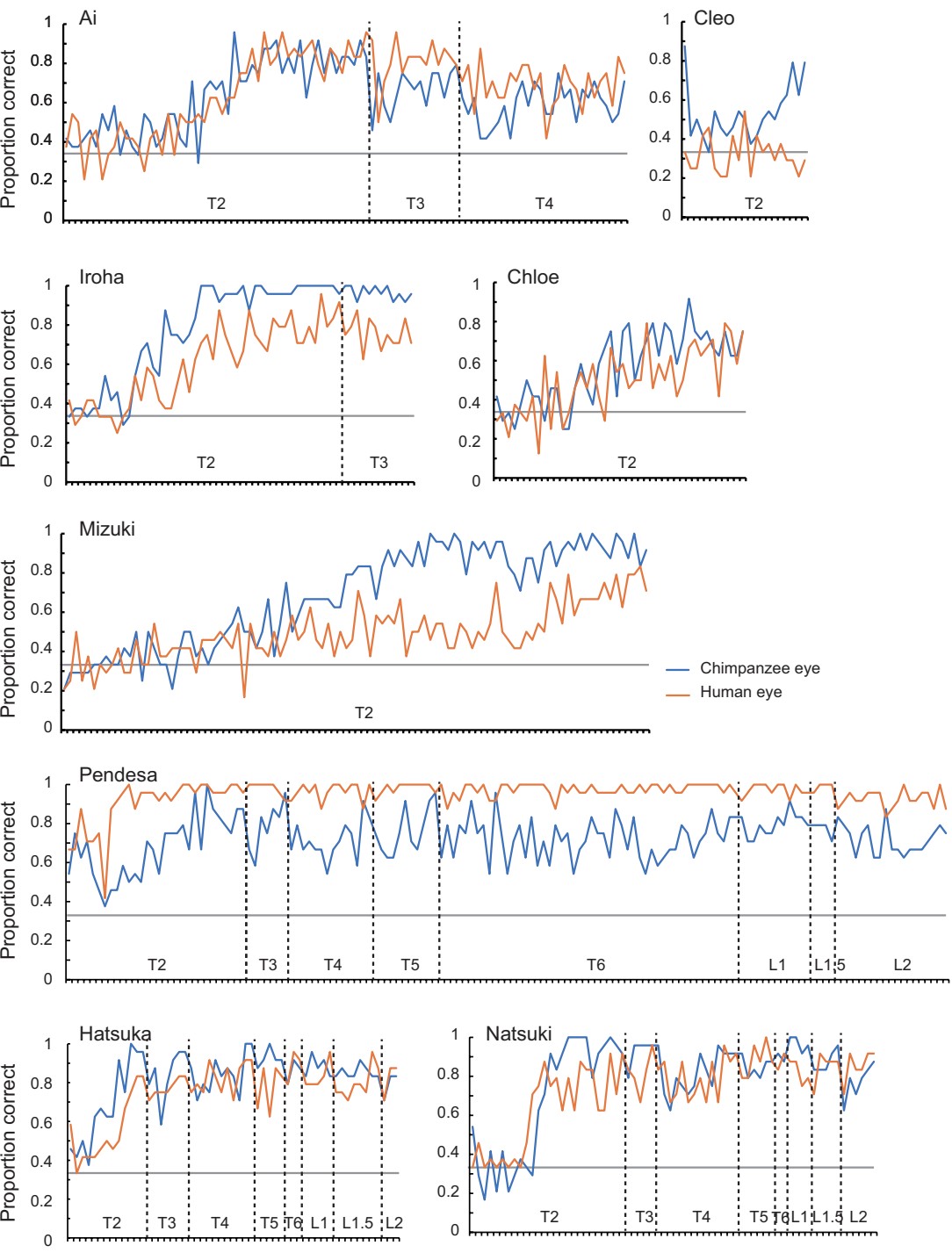

**Appendix 4—figure 1.** Performance of chimpanzee participants (Ai, Cleo, Chloe, Iroha, Mizuki, Natsuki, Hatsuka, and Pendesa) during training stages 2–6 (T1–6; T1 was not included in this graph because it was a pilot session presenting chimpanzees with the no-iris target stimuli) and pre-Test A1 sessions (L1–2), represented as raw proportion correct across sessions. See *Appendix 4—table 2* for the number of sessions required for each training and pre-Test A1 stage in each chimpanzee participant.

