## [Editor Report]

The study by Kano et al. is notable for being the first to adopt a comparative experimental approach, testing both humans and chimpanzees in a cross- and within-species design, demonstrating that uniformly white sclera enhances eye-gaze discrimination in both species. Crucially, their results support the gaze-signaling hypotheses for the evolution of particular features of the human eye but further suggest that uniformly white sclera are critical for eye-gaze discrimination when visibility conditions are poor.

---

## [Decision Letter]

**Decision letter after peer review:**

Thank you for submitting your article "Experimental evidence for the gaze-signaling hypothesis: White sclera enhances the visibility of eye- gaze direction in humans and chimpanzees" for consideration by *eLife*. Your article has been reviewed by 2 peer reviewers, including Ammie K Kalan as the Reviewing Editor and Reviewer #1, and the evaluation has been overseen by Detlef Weigel as the Senior Editor. The following individual involved in review of your submission has agreed to reveal their identity: Kai R Caspar (Reviewer #2).

Essential revisions:

Both reviewers found this study well executed with carefully designed experiments and transparent reporting of procedures. However a number of key issues require some attention during revisions:

1) Can you clarify and spend more time in the introduction explaining the general aim of your study followed by the justification and rationale for your experimental design given each one of your hypotheses. At the moment the hypotheses are encountered without a clear, overarching picture of the study's aims. Some hypotheses/tests are also not clearly explained. Why for example was inverted eye colour tested? (see also point 3).

2) Contextualize your results and findings better given that other nonhuman primates also have depigmented sclera and the fact that chimpanzees' dark sclera may not have been the ancestral condition (see Reviewer 2's detailed concerns). Can the authors spend more time to discuss potential alternative explanations given the variation observed in sclera colouration in other great apes?

3) Regarding your stimuli of inverting colours, the design procedure does not match the stimuli shown in Figure 1. If there was an inversion of iris and sclera color (including contrast and hue), then the inverted chimpanzee would need to have an amber-colored sclera and a blackish iris. Yet, the inverted chimp model apparently has a human-like white sclera and retains the amber-colored iris. Additionally, the pupil colors were also inverted for both species. Can you please explain this discrepancy and provide a clear explanation as to why these stimuli were edited in this manner, which hypotheses specifically they relate to, and whether you controlled for contrast and hue in this inverted/edited stimuli for the chimps given that the contrast values for the humans were equal in the two conditions. Please also provide additional examples of stimuli in the methods to aid clarification.

4) Clarify the GLMM code by adding the R script to aid transparency. It was not clear for example what the exact response variable was in your models.

5) Expand upon your limitations of the study section by explicitly stating the inability to generalize for chimpanzees given the low sample size and specific experience with touchscreens these chimps have, and the need for a cross cultural sample of human participants as well.

6) Clarify the differences between eye gaze and glancing/head direction with respect to communication and cooperation (see also Reviewer 2's comments re: conflation of these two concepts in the manuscript), and the potential implications/limitations of this for your experimental design and its interpretations.

*Reviewer #1 (Recommendations for the authors):*

Overall I find the study by Kano and colleagues will be a solid contribution to the literature on this important topic.

*Reviewer #2 (Recommendations for the authors):*

Introduction

There is some confusion in the primate cognition literature when it comes to the differentiation between glancing/eye gaze and gazing/head direction. From my impression, the authors are perfectly aware of this distinction and refer to it in several sections of the manuscript. Still, they do not adequately differentiate it in the text. For instance, in the first section of the introduction, they refer to "eye gaze" in humans, then switch to "gaze" in great apes without indicating that two fundamentally different signals are described. Later on, the authors equate "eye gaze" with "gaze" (l. 65) which should not be done to avoid further confusion. The clarity of the text would be significantly improved by briefly stating the differences between the two phenomena and how they are referred to throughout the text.

l. 52: Please note that the conjunctiva, which contribute to the dark-eyed phenotypes of many primate eyes, is also depigmented in humans.

l. 56: "eye shape" is an ambiguous expression. What exactly is meant here?

l. 64: perhaps replace "our" with "a".

l. 83: Please be precise when referring to "color". The studies you cite here did, for instance, not study hue, but exclusively relied on differences in gray values. In the supplements, you refer to the lab color space that I am unfamiliar with (and probably many readers as well). Does it allow a differentiation between (gray value) contrast and hue? Furthermore, chimpanzee eyes and human eyes indeed significantly deviate from each other in absolute irido-scleral contrast. They do not differ in relative iris luminance, which is way of iridoscleral contrast quantification that should not be applied for interspecific comparisons (see Caspar et al., 2021).

Figure 1: I'm not exactly sure if I understood what you mean by inverting eye color "without affecting the iris-sclera color contrast". The inverted chimpanzee figure does not look color inverted to me (shouldn't it have an amber-colored sclera and a black iris?).

Please also note that you inverted pupil color and explain why you did it. This decision leads to visibility of the pupil in the inverted chimp picture but not in the one of the inverted human. This might have implications for the experimental results. I suggest to at least briefly mention this issue.

Just out of curiosity: Is there a specific rationale behind the order in which you present the hypotheses? Why is the core hypothesis not #1?

Results

The study impressively demonstrates that chimpanzees have enormous difficulties to differentiate between glancing conditions, as shown by the extensive training needed to reliably accomplish this task and the high percentage of drop-outs. It has been prominently argued recently, that chimpanzee eyes are optimized to convey eye-gaze cues and that the role of glancing in chimpanzee social behavior has been underestimated (Perea-García et al., 2019). I believe it is important to point out how the new results challenge these proposals, which – if true – would bear important ramifications for chimpanzee communication.

You report only minor effects of eye shape – wouldn't this be expected, given the comparatively modest displacement of the iris relative to a direct glance condition is in your test stimuli?

Discussion

l. 237: Please note that critics of the gaze-signaling hypothesis do not deny that the human eye is more conspicuous than that of chimpanzees (Perea-García et al., 2019 are an exception). The key problem of this hypothesis is that it assumes a functional link between eye pigmentation and socio-cognition although there is no evidence for this. Humans are the only species in which this connection has been shown and it is not clear, whether this is a coincidental correlation. For instance, the hypothesis cannot explain extensive scleral depigmentation in non-human primates such as orangutans and howler monkeys. I would suggest to be more transparent when it comes to discussing recent criticisms of said hypothesis and how the results from this paper can add to the debate.

Methods

l. 430: "Training 431 stage 1 presented the target image with no iris with the distractor images with irises (in direct gaze)." Was pupil color affected as well? Perhaps an example of these test stimuli would be a good addition to the supplements, to give readers a better impression.

---

## [Author Response]

Essential revisions:Both reviewers found this study well executed with carefully designed experiments and transparent reporting of procedures. However a number of key issues require some attention during revisions:1) Can you clarify and spend more time in the introduction explaining the general aim of your study followed by the justification and rationale for your experimental design given each one of your hypotheses. At the moment the hypotheses are encountered without a clear, overarching picture of the study's aims. Some hypotheses/tests are also not clearly explained. Why for example was inverted eye colour tested? (see also point 3).

Accordingly, we substantially expanded our introduction section to better frame our hypotheses and experimental rationales. Please see our new introduction section (page 2-9).

Regarding the “inversion of eye color”, we actually should have said the “reversal of contrast polarity of eyes”, an experimental manipulation adopted in a related experimental study (Ricciardelli, et al., 2000). See below for the details about this point.

2) Contextualize your results and findings better given that other nonhuman primates also have depigmented sclera and the fact that chimpanzees' dark sclera may not have been the ancestral condition (see Reviewer 2's detailed concerns). Can the authors spend more time to discuss potential alternative explanations given the variation observed in sclera colouration in other great apes?

Accordingly, we also substantially expanded our Discussion section to better relate our results to those recent morphological studies on great ape eye color. Please see our new Discussion section (pages 15-18).

Also, in this revision, we can better clarify our hypotheses, rationales, and interpretations by referring to one most recent morphological study on great ape eye color (Kano, F., Furuichi, T., Hashimoto, C., Krupenye, C., Leinwand, J. G., Hopper, L. M.,... Tajima, T. 2021. What is unique about the human eye? Comparative image analysis on the external eye morphology of human and nonhuman great apes. Evolution and Human Behavior, in press, DOI: 10.1016/j.evolhumbehav.2021.12.004). Although our experimental study was independently performed from this related study, we have built some of our experimental designs based on the results from this related study. Yet, we could not refer to it extensively in our previous manuscript because this related study has not been out for a while (we apologize for any unclarity of our previous manuscript arising from this publication timing issue). The results from this related morphological study are complementary to those from this experimental study. We summarized the results from this related study in the introduction section, lines 91-104.

3) Regarding your stimuli of inverting colours, the design procedure does not match the stimuli shown in Figure 1. If there was an inversion of iris and sclera color (including contrast and hue), then the inverted chimpanzee would need to have an amber-colored sclera and a blackish iris. Yet, the inverted chimp model apparently has a human-like white sclera and retains the amber-colored iris. Additionally, the pupil colors were also inverted for both species. Can you please explain this discrepancy and provide a clear explanation as to why these stimuli were edited in this manner, which hypotheses specifically they relate to, and whether you controlled for contrast and hue in this inverted/edited stimuli for the chimps given that the contrast values for the humans were equal in the two conditions. Please also provide additional examples of stimuli in the methods to aid clarification.

We apologize for this confusion and our insufficient explanation for the rationales. Please note that we did not swap the color of the iris and sclera in the eye image but reversed the contrast polarity of the eye image (by inverting their grayscale values) following Ricciardelli et al., (2000). One advantage of our method is that we could change only contrast polarity but not any color differences within the eye image, namely the iris-sclera and pupil-iris color differences (thus this manipulation does not change the conspicuousness of iris or pupil *per se*). Please also note that the visibility of eye-gaze directions depends on the visibility of both iris and eye-outline edges, not that of only iris or pupil. To clarify this aspect, we added Figure S1.

To visually clarify this stimulus-making procedure, we added Figure S4.

We also added the complete set of chimpanzee stimuli in our online repository (https://osf.io/2xny3/?view_only=b03f2ee4adf549aea09d8df5ab88f126). Please understand that we cannot do this for our human stimuli due to the difficulty in obtaining permission to publish human faces online, but the original images can be seen in the published dataset (Eggar et al., 2011).

4) Clarify the GLMM code by adding the R script to aid transparency. It was not clear for example what the exact response variable was in your models.

Accordingly, we added Table S1 detailing R formulas.

5) Expand upon your limitations of the study section by explicitly stating the inability to generalize for chimpanzees given the low sample size and specific experience with touchscreens these chimps have, and the need for a cross cultural sample of human participants as well.

Accordingly, we added a paragraph stating the limitation of our study. Please see lines 323-337.

We clarified in our method section that the results from two related experimental studies (Ricciardelli et al., 2000; Yorzinski and Miller, 2020) are complementary to our results because these studies primarily tested human participants from other cultures (lines 105-120).

6) Clarify the differences between eye gaze and glancing/head direction with respect to communication and cooperation (see also Reviewer 2's comments re: conflation of these two concepts in the manuscript), and the potential implications/limitations of this for your experimental design and its interpretations.

We added this explanation in the first paragraph of our introduction section (lines 56-60).

Thank you very much for this helpful summary.

Reviewer #2 (Recommendations for the authors):IntroductionThere is some confusion in the primate cognition literature when it comes to the differentiation between glancing/eye gaze and gazing/head direction. From my impression, the authors are perfectly aware of this distinction and refer to it in several sections of the manuscript. Still, they do not adequately differentiate it in the text. For instance, in the first section of the introduction, they refer to "eye gaze" in humans, then switch to "gaze" in great apes without indicating that two fundamentally different signals are described. Later on, the authors equate "eye gaze" with "gaze" (l. 65) which should not be done to avoid further confusion. The clarity of the text would be significantly improved by briefly stating the differences between the two phenomena and how they are referred to throughout the text.

Accordingly, in the first paragraph of our introduction, we clarified this point (lines 56-58).

l. 52: Please note that the conjunctiva, which contribute to the dark-eyed phenotypes of many primate eyes, is also depigmented in humans.

Noted, although we do not have this part anymore in our revision.

l. 56: "eye shape" is an ambiguous expression. What exactly is meant here?

Accordingly, we detailed this in our expanded introduction section (lines 83-104).

l. 64: perhaps replace "our" with "a".

Addressed.

l. 83: Please be precise when referring to "color". The studies you cite here did, for instance, not study hue, but exclusively relied on differences in gray values. In the supplements, you refer to the lab color space that I am unfamiliar with (and probably many readers as well). Does it allow a differentiation between (gray value) contrast and hue? Furthermore, chimpanzee eyes and human eyes indeed significantly deviate from each other in absolute irido-scleral contrast. They do not differ in relative iris luminance, which is way of iridoscleral contrast quantification that should not be applied for interspecific comparisons (see Caspar et al., 2021).

We detailed CIELAB color space in lines 464-467. Kano et al., (in press) used this color space to code both lightness (grayscale) and chromaticity components of great ape eye colors. This current study employed the same method to code the eye colors of stimuli. CIELAB color space is designed as a perceptual uniform color space for humans and is considered applicable to nonhuman primates with human-like trichromatic vision.

Figure 1: I'm not exactly sure if I understood what you mean by inverting eye color "without affecting the iris-sclera color contrast". The inverted chimpanzee figure does not look color inverted to me (shouldn't it have an amber-colored sclera and a black iris?).Please also note that you inverted pupil color and explain why you did it. This decision leads to visibility of the pupil in the inverted chimp picture but not in the one of the inverted human. This might have implications for the experimental results. I suggest to at least briefly mention this issue.

As noted earlier, in our stimulus manipulation, we reversed the contrast polarity of the eye images but did not just swap the iris and sclera colors. Please note that the reversal of the contrast polarity of the eye image does not change the iris-sclera and iris-pupil color differences (lines 144-146) and thus the conspicuousness of iris or pupil *per se*.

We added Figure S4 to clarify our stimulus-making procedures.

Just out of curiosity: Is there a specific rationale behind the order in which you present the hypotheses? Why is the core hypothesis not #1?

True. It should be easier to understand in that way. Addressed.

ResultsThe study impressively demonstrates that chimpanzees have enormous difficulties to differentiate between glancing conditions, as shown by the extensive training needed to reliably accomplish this task and the high percentage of drop-outs. It has been prominently argued recently, that chimpanzee eyes are optimized to convey eye-gaze cues and that the role of glancing in chimpanzee social behavior has been underestimated (Perea-García et al., 2019). I believe it is important to point out how the new results challenge these proposals, which – if true – would bear important ramifications for chimpanzee communication.

Kano et al., (in press) commented on Perea-Garcia (2019). Briefly, this morphological study found that the eye regions of nonhuman great apes are as salient as those of humans *within* the face (mainly due to the iris-sclera color difference). However, due to the lack of clarity of eye-outline edges in the nonhuman eye (mainly due to dark or graded/patchy sclera colors), their eye-gaze directional cues are less clearly visible compared to those of humans. We summarized these findings in our introduction section (lines 91-103).

You report only minor effects of eye shape – wouldn't this be expected, given the comparatively modest displacement of the iris relative to a direct glance condition is in your test stimuli?

This is a good point. However, previous eye-tracking studies measuring both species’ eye movements under naturalistic conditions reported that the majority of eye positions fall within 20 degrees in both species (Kothari et al., 2020; Kano and Tomonaga, 2013). We mentioned this in lines 289-295.

Discussionl. 237: Please note that critics of the gaze-signaling hypothesis do not deny that the human eye is more conspicuous than that of chimpanzees (Perea-García et al., 2019 are an exception). The key problem of this hypothesis is that it assumes a functional link between eye pigmentation and socio-cognition although there is no evidence for this. Humans are the only species in which this connection has been shown and it is not clear, whether this is a coincidental correlation. For instance, the hypothesis cannot explain extensive scleral depigmentation in non-human primates such as orangutans and howler monkeys. I would suggest to be more transparent when it comes to discussing recent criticisms of said hypothesis and how the results from this paper can add to the debate.

This is also a very good point. As noted earlier, we clarified this in the limitation section of our discussion (line 323-337).

Methodsl. 430: "Training 431 stage 1 presented the target image with no iris with the distractor images with irises (in direct gaze)." Was pupil color affected as well? Perhaps an example of these test stimuli would be a good addition to the supplements, to give readers a better impression.

Accordingly, we added the complete set of our chimpanzee stimuli in our online repository (https://osf.io/2xny3/?view_only=b03f2ee4adf549aea09d8df5ab88f126), and moreover also the examples of our training stimuli in Figure S5.